



# An urban large-eddy-based dispersion model for marginal grid resolutions: CAIRDIO v1.0

Michael Weger[1], Oswald Knoth[1], and Bernd Heinold[1]

[1]Leibniz Institute for Tropospheric Research, Leipzig, Germany

**Correspondence:** Michael Weger (weger@tropos.de)

**Abstract.** The capability for high spatial resolutions is an important feature of accurate numerical models dedicated to simulate the large spatial variability of urban air pollution. On the one hand, the well established mesoscale chemistry transport models have their obvious short-comings attributed to their extensive use of parameterizations. On the other hand, obstacle resolving computational fluid dynamic models, while accurate, still often demand too high computational costs to be applied on a regular

basis for entire cities. The major reason for the inflated numerical costs is the required horizontal resolution to meaningfully apply the obstacle discretization, which is most often based on boundary-fitted grids, like e.g. the marker-and-cell method. Here we present a large-eddy-simulation approach that uses diffusive obstacle boundaries, which are derived from a simplified diffusive interface approach for moving obstacles. The diffusive interface approach is well established in two-phase modeling, but to the author's knowledge has not been applied in urban boundary layer simulations so far. Our dispersion model is capable

of representing buildings over a wide range of spatial resolutions, including grid spacings equal or larger than typical building size. This opens up a very promising opportunity for application of accurate air quality simulations and forecasts on entire mid-sized city domains. Furthermore, our approach is capable of incorporating the influence of the land orography by the additional optional use of terrain-following coordinates. We validated the dynamic core against a set of numerical benchmarks and a standard high-quality wind-tunnel data set for dispersion-model evaluation.

## 1 Introduction

The state-of-the-art in urban air quality modeling now almost routinely incorporates the scales at which processes governing the atmospheric dispersion inside the urban planetary boundary layer (PBL) are explicitly represented (Benavides et al., 2019; Croitoru and Nastase, 2018; Kadaverugu et al., 2019). The reasons for this increase in physical detail are manifold. On the one hand, even if PBL mixing processes can be parameterized, the parameterizations itself must rely on a sound physical basis for

which often detailed large-eddy simulations (LES) are consulted (Noh and Raasch, 2003; Kanda et al., 2013). A direct benefit of more detailed numerical simulations is the ability to provide high-resolution urban air-quality data for research purposes, like studying source attribution (Fernández et al., 2019), exposure risk assessment (Chang, 2016), and eventually to provide more representative forecasts for individual locations (Carlino et al., 2016). Exposure relevant pedestrian-level pollution concentrations are subjected to a considerable and also complex spatiotemporal variability, as they are not only influenced by the

relative location of pollution sources, but also very importantly by the urban morphology and the interacting meteorological



conditions (Birmili et al., 2013; Paas et al., 2016; Harrison, 2018). For an accurate simulation, it is thus not only key to explicitly represent the urban canopy features, but also to consider the mesoscale prevailing meteorological conditions surrounding cities. Depending on the level of physical detail, a trade-off in the use of high-resolution numerical simulations often remains their exclusive limited-area applicability, which can be very restrictive. As a result, an active topic of research is dedicated to the increase in the numerical efficiency of high-resolution modeling tools and their incorporation into a larger modeling framework (Baik et al., 2009; Jensen et al., 2017; Kurppa et al., 2020).

Commonly used multiscale approaches consist of nested domains and involve different types of models designed for a specific scale range. To address the global and regional scales, the use of chemistry transport models (CTM) in combination with numerical weather prediction (NWP) models is a well established practise. Examples of such model combinations include, C-IFS (Flemming et al., 2015), WRF-Chem (Grell et al., 2005), CMAQ (Appel et al., 2017), ICON-ART (Rieger et al., 2015) and COSMO-MUSCAT (Wolke et al., 2004, 2012). In all of these models, subgrid-scale effects of temperature and moisture, as well as PBL dynamics are parameterized. The influence of the urban canopy on the PBL in NWP models can be considered through sophisticated canopy parameterizations (Martilli et al., 2002; Schubert et al., 2012). The improvements of these so-called urbanized NWP's substantially reflect in modeled pollutant concentrations (Kim et al., 2015; Wang et al., 2019). Nevertheless, as the model domain contains no explicit representation of buildings, the pollutants emitted at street level are diluted throughout the entire grid cell, which can considerably deviate from real conditions, where a large part of the physical volume may be inaccessible. As a result, pollutant concentrations modeled using NWP approaches are more representative to the urban back-ground (Korhonen et al., 2019). Another practical resolution and hitherto accuracy limit of NWP models results from the use of parameterizations itself. Using WRF, Haupt et al. (2019) observed that for horizontal grid spacings below the typical height of the PBL numerical results can become spurious. Based on their findings, they recommend not to apply NWP on the sub-kilometer scale without careful replacement of the used parameterizations. In PBL meteorology, the microscale follows seamless at the lower limit of the mesoscale (super-kilometer range) (Stull, 1988; Rakai and Gergely, 2013). However, adopting the modeling perspective, there is a clear segregation of the microscale from the mesoscale. While the latter is truncated at the lower resolution end through the extensive use of parameterizations, the attempt to model urban PBL processes directly using microscale approaches requires an adequate spatial resolution (few tens of meters). The landscape of microscale models is diverse (Fallah-Shorshani et al., 2017; Brown, 2014; Hanna et al., 2006) and it reflects the difficulty in trading off computational resources and accuracy. The microscale pendants of NWPs can be considered as the computational fluid dynamics (CFD) models, among them LES approaches are the most accurate but expensive ones. LES models resolve the turbulent spectra up to the filter cut-off size (often the grid size) and rely on simplistic sub-scale parameterizations only (Maronga et al., 2019). These two different approaches (extensive parameterization vs. explicit representation) are difficult to merge at the bridging scale-range (few tens of meters to one kilometer), for which reason Haupt et al. (2019) introduced the key word "terra-incognita". An exemplary study where LES have been attempted at horizontal resolutions up to above $100\,\mathrm{m}$ is given by Efstathiou et al. (2016). However, their simulations did not include an urban canopy, whose discretization would impose a much stringend resolution limit. In fact, obstacle-resolving urban LES (ULES) studies are typically performed at spatial resolutions of less than $10\,\mathrm{m}$ to $20\,\mathrm{m}$. Even at such comparatively coarse resolutions, computational resources are still





the limiting factor as they restrict ULES applications to research purposes or to areas encompassing city parts only (Wolf et al., 2020). Further depending on the degree of physical detail (e.g. costly radiative transfer calculations in a complex urban environment), ULES have not yet become feasible choices to simulate air-quality in cities on a regular basis, despite their obvious advantages concerning physical accuracy.

In physical computing, it is well known that only a slight increase in the grid spacing results in a large computational saving (e.g. models using explicit time integration are ideally integrated 16 times faster on the same domain using a grid with doubled grid spacings in all 3 dimensions). The savings in turn could be invested in larger physical domains for more holistic simulations. The additional domain extend would also provide an additional relaxation fetch for the mesoscale forcing in case complete cities are attempted to be modeled. As a result of an increased grid spacing, the level of detail of modeled wind
fields clearly decreases. Nevertheless, the effect of a lower spatial resolution on the accuracy of dispersion simulations can be expected to be less detrimental in practical modeling examples, as it is affected also by other major uncertainties, like e.g. emission distribution.

The key feature which sets the technical limit to the model resolution of ULES applications is clearly the spatial discretization of obstacles. A class of methods in which the obstacle geometry is not tied to the grid are the immersed boundary methods,
summarized by Mittal and Iaccarino (2005). These are essentially Cartesian methods, but instead of relying on a commonly used marker-and-cell method (grid cells are assigned either to the building interiors or to the atmosphere), they aim at representing the boundary condition associated with a rigid boundary (e.g. Neumann for pressure) on grid cell faces not coinciding with the obstacle boundary. Among these methods, the so-called direct forcing uses the ghost-cell interpolation technique, where image points from adjacent interior ghost points are mirrored across the rigid boundary and interpolated using surrounding
fluid nodes. While this method greatly enhances the flexibility in the choice of the grid size, it still suffers from the empirical nature in the selection of the interpolation nodes and the interpolation method itself. On the other hand, an equivalent boundary forcing can be more rigorously deduced from a two-phase flow model (Drew, 1983) when neglecting the restoring source terms. Treating the second phase as an indeformable solid, Kemm et al. (2020) derived a diffusive-interface (DI) model for compressible fluids. One of the main advantages of their approach is the algorithmic simplicity, as the boundary-forcing term
is analytically coupled to the DI, which is advected as a scalar.

In this work, we represent a new LES-based dispersion model as a computationally feasible yet accurate downscaling tool for mesoscale meteorology and air pollution fields over urban areas. The development adopts the basic idea of DI, as it is simple and efficient to implement and provides a high flexibility in the choice of the spatial resolution. As the city-structure is assumed to be static, DI simplifies to static diffusive obstacle boundaries (DOB). By considering the conservation law in a finite-volume
framework, DOB is incorporated in the discrete differential operators, making the governing equations appear in their familiar form. The LES equations can then be solved on a structured Cartesian grid with standard methods. To interpret our DOB approach from a physical point of view, it can be argued that the grid cells are interspersed with a porous and semi-permeable medium, whose detailed structure is only of concern insofar as to which degree it modifies the mass and momentum budged at the grid level through two different types of interface fields. To the authors knowledge, this approach has not been used for
air quality modeling so far, but very interestingly the concept of permeability finds application in geological science (Haga,





2011). In contrast to geometry-aligned discretizations, which preserve a high degree of accuracy near obstacle walls but require high resolutions, this approach suits more to the integral aspect of building shapes and configurations at marginal resolutions. However, by increasing the grid resolution, the approach transforms seamless into a traditional obstacle resolving Cartesian approach, as the interface eventually becomes sharply defined and imposes Neumann-boundary conditions on the pressure.

The paper is organized as follows: Section 2 provides a detailed model description, including the spatial discretization. Section 3 is dedicated to the evaluation of the model using idealized test cases and a more complex and realistic wind-tunnel dispersion experiment. Finally, in Section 4 the benefits and limitations of the approach are summarized and concluded by an outlook for potential future applications.

## 2  Model description

### 2.1  Basic equations

The physical domain consists of an interspersed, stationary solid phase representing the buildings, and a mobile fluid phase. The governing equations for the mobile phase are deduced from a simplified two-phase model neglecting restoring forces. In a two-phase model, phase-fraction functions $\alpha_1$, $\alpha_2$ with $0 \leq \alpha_1 \leq 1$ and $\alpha_1 + \alpha_2 = 1$ are used to formulate the set of equations of both phases individually. The equation of motion of an incompressible phase is adopted from Drew (1983) (eq. 41, 45

therein). By setting the interfacial force density and surface tension to zero, the simplified momentum equation of the mobile phase (indicated with $\alpha_1$) is written as:

$$\partial_t \left( \alpha_1 \rho \boldsymbol{u} \right) = -\nabla \cdot \left( \alpha_1 \rho \boldsymbol{u} \otimes \boldsymbol{u} \right) - \alpha_1 \nabla \left( \alpha_1 p \right) + p_I \nabla \alpha_1 + \nabla \cdot \left( \alpha_1 \mathbf{T} \right) + \alpha_1 \rho \boldsymbol{b} \tag{1}$$

$p_I$ is the interface pressure, reflecting Newton's third law of motion near a fixed wall. As in Kemm et al. (2020), it will be assumed that it is in equilibrium with the surrounding fluid pressure $p$. $\mathbf{T}$ is the stress tensor, which in LES averaging

contains the contributions from subscale and surface fluxes. In this model, the implicit approach is used for spatial filtering (Schumann, 1975). Viscous stresses are neglected due to the high Reynolds numbers typically encountered in atmospheric flows. $\boldsymbol{b}$ corresponds to the sum of external body forces, which also contains inertial forces in a rotating frame of reference. The interface in our case is static, as buildings are assumed not to be affected by the flow. This makes it possible to multiply the equation of motion with $\alpha_1^{-1} \rho^{-1}$, to obtain the tendency equation in single-flow denotation. The density is not allowed to

change in time and is denoted with $\rho_{ref}$ for the reference state, which makes sense for an incompressible fluid. The full set of model equations reads:

$$\partial_t \boldsymbol{u} = -\alpha_1^{-1} \nabla \left( \alpha_1 \boldsymbol{u} \otimes \boldsymbol{u} \right) - \frac{1}{\rho_{ref}} \alpha_1^{-1} \alpha_1 \nabla p - \alpha_1^{-1} \nabla \cdot \left( \alpha_1 \mathbf{T} \right) - \boldsymbol{f} \times \boldsymbol{u} + \boldsymbol{g} \frac{\overline{\Theta}_v - \Theta_v}{\overline{\Theta}_v} \tag{2}$$

$$\alpha_1^{-1} \nabla \cdot \left( \alpha_1 \boldsymbol{u} \right) = 0. \tag{3}$$





$$\partial_t \Theta = -\alpha_1^{-1} \nabla \cdot (\alpha_1 \boldsymbol{u} \Theta) + \alpha_1^{-1} \nabla \cdot (\alpha_1 k_h \nabla \Theta) + \alpha_1^{-1} S_\Theta \tag{4}$$

$$\partial_t Q_v = -\alpha_1^{-1} \nabla \cdot (\alpha_1 \boldsymbol{u} Q_v) + \alpha_1^{-1} \nabla \cdot (\alpha_1 k_h \nabla Q_v) + \alpha_1^{-1} S_{Q_v} \tag{5}$$

$$\Theta_v = \Theta (1 + 0.61 Q_v) \tag{6}$$

In the momentum equation (eq. 2), the body-force term was replaced by the Coriolis term, containing the Coriolis parameter $\boldsymbol{f}$ for a mean geographic latitude, and with the buoyancy term using the Boussinesq approximation. $\boldsymbol{g}$ is the gravity-acceleration vector in the local frame of reference, $\Theta_v$ the virtual potential temperature defined by Eq. 6, and the overbar denotes for the

horizontal mean state. Eq. 3 is the continuity equation derived by the same arguments from the original equation in Drew (1983). The transport equation for scalars, like $\Theta_v$ and specific humidity $Q_v$, contains source terms from parameterized surfaces fluxes. These have to be multiplied with $\alpha_1^{-1}$. The scalar field $k_h$ is the eddy-diffusion for heat, which arises from the spatial filtering. Note that in our case, the combinations $\alpha_1^{-1} \nabla \cdot \alpha_1$ and $\alpha_1^{-1} \alpha_1 \nabla$ can be identified as particular versions of the divergence and gradient operator, which incorporate diffusive boundaries. It is also interesting to mention that the stencil of $\alpha_1$ (face-centered)

differs from $\alpha_1^{-1}$ (volume-centered) on a staggered grid.

## 2.2 Computation grid and diffuse boundaries

The computation uses a structured hexaedral grid, with the option for local grid stretching in all 3 dimensions. The velocity components are defined at the cell faces, whereas scalar fields are cell centered, classifying the grid structure as Arakawa-C type. Vertical coordinate transformation allows for a curve-linear grid in the physical space, which can be used to follow a

smoothly varying terrain function:

$$\tilde{x} = x, \quad \tilde{y} = y, \quad \tilde{z} = z - h(x, y), \tag{7}$$

$z$ is the mean-sea level height and $h(x, y)$ the terrain-height function. The first grid level matches the terrain function, and all elevated levels are given by adding a constant vertical increment to the first level.

The pressure gradient in terrain-following coordinates expands to:

$$\nabla p = \tilde{\nabla} p - [\partial_x h(x, y) + \partial_y h(x, y)] \partial_z p \tag{8}$$





The horizontal derivatives of $h(x,y)$ are discretized with second-order differences and linearly interpolated on the respective cell-faces, where needed.

For the divergence operator, the contra-variant velocity components which are normal to the cell faces are needed. The horizontal components under the vertical coordinate transformation are indifferent, and the vertical contra-variant velocity is given by:

$$\omega = w - \partial_x h u - \partial_y h v. \tag{9}$$

Using Eq. 9, the advective tendency of a scalar $q$ can be written as:

$$\partial_t q = -u\partial_x q - v\partial_y q - [w - u\partial_x h - v\partial_y h]\partial_z q \tag{10}$$

Similarly, the divergence-free criterion follows in terrain coordinates:

$$\partial_x u + \partial_y v + \partial_z (w - \partial_x h u - \partial_y h v) = 0 \tag{11}$$

Combining Eq. 11 and 8, the metric terms in the Laplace operator are obtained, which is needed in the pressure equation:

$$\triangle = \tilde{\triangle} - \partial_x (\partial_x h \partial_z) - \partial_y (\partial_y h \partial_z) - \partial_z (\partial_x h \partial_x) \tag{12}$$
$$- \partial_z (\partial_y h \partial_y) + \partial_z \left[(\partial_x h)^2 \partial_z\right] + \partial_z \left[(\partial_y h)^2 \partial_z\right]$$

As $u$, $v$, and $w$ are maintained as the prognostic model variables and $\boldsymbol{g}$ is invariant under the given coordinate transformation, no metric terms arise in the buoyancy term. However, the horizontal averaging requires a remapping of $\Theta_v$ from $\tilde{z} = const$ to $z = const$.

The spatial discretization uses a finite volume method, which allows a consistent treatment of the diffusive obstacle interface. The conservation of a scalar $q$ under phase partitioning is formulated using the Gauss theorem:

$$\int_{\Delta V} \partial_t \left[q_m \chi_m + q_s (1 - \chi_m)\right] dV = \tag{13}$$
$$- \int_{\partial(\Delta V)} \left[\boldsymbol{\eta_m} \cdot \boldsymbol{u_m} q_m + (1 - \boldsymbol{\eta_m}) \cdot \boldsymbol{u_s} q_s\right] dA$$

Here, subscript m refers to the mobile phase and subscript s to the solid phase for the building-interior. $\chi_m$ is the volume-
fraction function of the mobile phase and $\boldsymbol{\eta}_m$ the area-fraction function over which the flux of the mobile phase is integrated.




As already mentioned, the stationary solid phase is dropped, as it is $\partial_t\chi_m = 0$, $\partial_t q_s = 0$ and $\boldsymbol{u_s} = 0$. This simplifies the conservation form and finally leads to the particular differential form of the transport equation for the mobile phase:

$$\partial_t q_m = -\frac{1}{\chi_m \Delta V} \int_{\partial(\Delta V)} \boldsymbol{\eta_m} \cdot \boldsymbol{u_m} q_m \, \mathrm{d}A = -\frac{1}{\chi_m} \nabla \cdot \boldsymbol{\eta}_m (\boldsymbol{u_m} q_m) =: -\nabla_m \cdot (\boldsymbol{u_m} q_m) \tag{14}$$

The subscript $m$ was only briefly introduced here and will be dropped again, as only one phase, the mobile phase, is of
interest.

Using the Cartesian grid structure, the flux-divergence can be put in a discrete form:

$$
\begin{aligned}
\nabla \cdot \boldsymbol{f} = \frac{1}{\chi \Delta V} \big[ & (\eta_x \Delta A_x f_x)^L - (\eta_x \Delta A_x f_x)^R \\
+ & (\eta_y \Delta A_y f_y)^L - (\eta_y \Delta A_y f_y)^R \\
+ & (\eta_z \Delta A_z f_z)^L - (\eta_z \Delta A_z f_z)^R \big]
\end{aligned}
\tag{15}
$$

$\Delta V$ and $\Delta A_{x,y,z}$ are the cuboid volumes and face areas, respectively. The superscripts R and L refer to the left and right cell face for each dimension, respectively. Note that for $f_z$, the contra-variant flux has to be used.

The pressure gradient components on the cell faces are discretized with second-order accuracy. For example, the gradient
components on the x-orientated cell faces, and z-orientated faces are:

$$
\begin{aligned}
\partial_x p = & \frac{2\eta_x \Delta A_x}{(\chi \Delta V)^L + (\chi \Delta V)^R} \left( p^R - p^L \right) \\
& - \frac{1}{\Delta_x} \left( h^R - h^L \right) \mathbf{L}^{z \to x} \partial_z p,
\end{aligned}
\tag{16}
$$

$$\partial_z p = \frac{2\eta_z \Delta A_z}{(\chi \Delta V)^L + (\chi \Delta V)^R} \left( p^R - p^L \right), \tag{17}$$

where $\mathbf{L}^{z \to x}$ is the linear interpolation operator from a z-face to an x-face, and $\Delta_x$ is the cuboid size used for differencing
the terrain function.

For a grid-conforming surface, the area fractions $\eta_{x,y,z} = 0$, and it follows that the required Neumann-boundary condition is satisfied. Furthermore, if a grid cell is surrounded by grid-conforming surfaces, the pressure value inside is decoupled from any exterior grid cell, which matches the physical meaninglessness of such a value. For partially open semi-permeable cell-faces the boundary condition is imposed only on a fraction of the cell face area.






The scaling fields for the cell volumes and face areas intersected by buildings are derived from geometric building data sets. As an alternative to terrain-following coordinates, it is possible to use diffuse boundaries for the terrain, in which case, the subsurface is represented by an obstacle too. Per definition, the volume-fraction field $\chi$ is expressed as the fraction of the building-free volume in each grid cell. For numerical reasons, however, $\chi$ is limited to a value of about $0.01$. For well resolved

buildings, the area-scaling fields $\eta_{x,y,z}$ would be derived by calculating the intersections of the buildings with the grid-cell faces. For under-resolved buildings, this method is very sensitive to the grid alignment, and in certain cases, buildings are missed, if they do not intersect with the cell faces, but nevertheless block the flow. Figure 1a) shows two possible scenarios, where grid cells are intersected by a building, which obviously blocks the flow in one dimension entirely but does not intersect with the cell faces of the blocked flow direction. In order to reliably capture such bottlenecks, a modified method to calculate

the area fractions is used. The grid-cell volume is partitioned in slices, with the slicing planes displaced along the dimension considered (e.g. $x-$dimension for the $yz-$faces). The minimum value of the free-volume fractions of these slices defines a so-called cell-area-scaling factor, which is assigned to the face being closer to the obstacle. For a more robust capture of non-parallel building walls, the slicing can be repeated several times with a slight rotation of the plane normal. If some cell faces are assigned to both values from the adjacent left and right grid cell, respectively, the minimum of both values is taken. For

the cell-faces left without assignment, the geometric intersection with buildings is calculated as suitable for the resolved case. Figure 1b shows on a practical example that the method preserves the blocking effect of a triangular-shaped building, even at spatial resolutions, which would be too coarse to resolve the building walls.

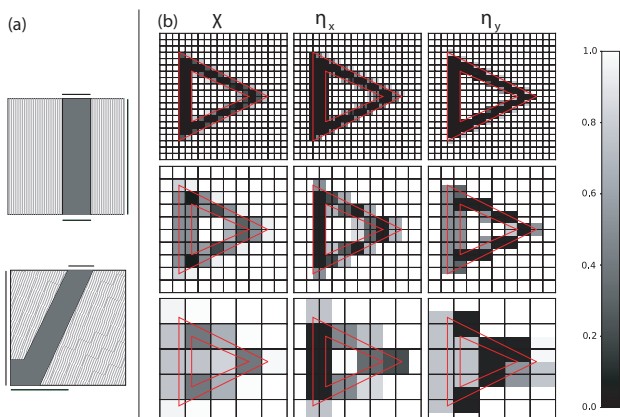

**Figure 1.** (a) Depiction of two different building sections (gray filled areas) inside a grid cell. The black lines mark the effectively blocked area of the respective side. The line pattern symbolises the slicing of the grid cell volume, which is here shown only for the x-dimension. (b) Scaling factors for a complete building, now depicted as fields calculated for 3 different grid resolutions. $\chi$ is the volume-scaling field, and $\eta_x$, $\eta_y$ are the horizontal components of the area-scaling field.





## 2.3 Advection scheme

The main task in the discretization of the advection term in a finite-volume framework is the reconstruction of the cell-averaged

fluxes on the faces to calculate the balance according to Eq. 15. It is noted, however, that even in the incompressible case, the

flux-balance form differs from the required advective form, as there is a residual divergence originating from the approximate

solution of the pressure equation.

$$(\partial_t q)_{adv} = -\nabla \cdot (\boldsymbol{u} q) + q \nabla \cdot \boldsymbol{u} \tag{18}$$

The reconstruction for a scalar reduces to several high-order 1-dimensional reconstructions of the cell-averaged value $q$ on

the cell faces. The fluxes are then obtained by multiplication with the exact momentum components, and are of the same spatial

accuracy as the reconstructed scalars. For the reconstruction itself, upwind-biased stencils are used, requiring two reconstruc-

tions on each cell face for the two flow directions. Since the grid-spacing is not equidistant, the reconstruction coefficients have

to be calculated in advance of the simulation as follows: Requiring the reconstruction to be $n^{th}$-order accurate, $n$ cell averaged

values are needed. Assuming the wind is blowing from the left (positive direction), the set of values encompasses $q(j-r+s)$,

$s \in [0, n-1]$, with $n \mod 2 = 1$, $r = (n-1)/2 + 1$ and $j$ being the target cell-index for the flux-balance calculation. For the

negative direction, it is $r = (n-1)/2$. In total, $n+1$ different values are needed.

The reconstruction itself is performed using the spatial derivative of the Lagrange polynomial, which fits the primitive

function evaluated at the cell faces (see e.g. Shu, 1998):

$$P(x) = \sum_{s=0}^{n-1} q(j-r+s)\Delta_x(j-r+s) \sum_{m=s+1}^{n} \frac{\sum_{l=0, l \neq m}^{n} \prod_{p=0, p \neq m, l}^{n} (x - x_{j-r+p-1/2})}{\prod_{l=0, l \neq m}^{n} (x_{j-r+m-1/2} - x_{j-r+l-1/2})} \tag{19}$$

The coefficients have to be evaluated at the cell faces $j-1/2$ for the positive upwind direction, and $j+1/2$ for the negative

one. The differences in parenthesis can be expressed in terms of the grid spacing. Instead of the geometric grid spacing, the ef-

fective grid spacing $\Delta_x^{eff}$, which is derived from the scaling fields, is used as a pseudo grid. This ensures a decreased weighting

of interpolation nodes within volume-compromised cells and prevents the scheme from interpolating the degenerated values

inside closed cells.


$$\Delta_x^{eff} = \frac{2\chi}{\eta_x^L + \eta_x^R} \Delta_x \tag{20}$$

$$x_{j-1/2} - x_{j-r+p-1/2} =: \tag{21}$$

$$\begin{cases} \sum_{l=0}^{r-p-1} \Delta_x^{eff}(j-r+p+l) & r-q \geqslant 0 \\ \sum_{l=0}^{p-r-1} -\Delta_x^{eff}(j+l) & p-r > 0, \end{cases}$$





and

$$x_{j+1/2} - x_{j-r+p-1/2} =: \tag{22}$$


$$\begin{cases} \sum_{l=0}^{r-p} \Delta_x^{eff}(j-r+p+l) & r-p+1 \geqslant 0 \\ \sum_{l=1}^{p-r-1} -\Delta_x^{eff}(j+l) & p-r-1 > 0. \end{cases}$$

While the formulas permit reconstructions of arbitrary order, a $5^{th}$-order reconstruction was found to give superior resolution to a $3^{rd}$-order one, while still being computationally efficient. For any increase in the order, additional ghost layers need to be communicated. Additionally, limiting of the reconstructions is applied using a total-variation diminishing method of Sweby

(1984). This will ensure monotonicity for scalar advection.

The two limited reconstructions $q^+$ and $q^-$ are finally combined to give the numerical flux based on the propagation velocity $u$:

$$(uq) = 0.5 \left[ u(q^+ + q^-) - ||u||(q^+ - q^-) \right] \tag{23}$$

For momentum transport, the routine for scalar cell-centered advection is re-used to advect a left-faced $u^l$ and right-faced

value $u^r$ for each momentum component (Hicken et al., 2005; Jähn et al., 2015), resulting in a total of 6 routine calls. In contrast to scalar advection, monotonicity is generally not desired, as it has a detrimental impact on energy conservation and will interact with the subgrid model. To prevent wiggles near building walls and to increase numerical stability, only local limiting is applied at obstacle boundaries. For a reconstruction face at index position $j$ located near a diffusive boundary, a local weight can be taken as $\alpha_x^+ = |(\eta_x(j-1) - \eta_x(j)|$ for the positive upwind direction, and $\alpha_x^- = |(\eta_x(j+1) - \eta_x(j)|$ for

the negative direction. If $u_{Lim}^+$ is the limited reconstruction of $u^+$, using the aforementioned weight $\alpha_x^+$, the following merging results in a weighted limiting of the velocity component perpendicular to an adjacent building wall:

$$u_{Limobs}^+ = (1 - \alpha_x^+)u^+ + \alpha_x^+ u_{Lim}^+ \tag{24}$$

The scheme, if necessary, degrades only to first-order accuracy at obstacle boundaries, while in the free boundary layer, no additional numerical dissipation is introduced.


The final momentum tendencies are obtained by interpolation of two centered tendencies on the face:

$$\partial_t^{adv} u = \frac{\left(\chi \Delta V \partial_t^{adv} u^l\right)^R + \left(\chi \Delta V \partial_t^{adv} u^r\right)^L}{(\chi \Delta V)^R + (\chi \Delta V)^L} \tag{25}$$

Note that it is combined the right-faced advective tendency from the left grid cell and the left-faced tendency from the right grid cell (superscripts L and R, respectively). Spatial accuracy of momentum advection is limited to the order of this

interpolation procedure, which is of second order.





## 2.4 Model integration, pressure correction method, and lateral boundary conditions

In the incompressible-flow equations, the pressure gradient term is not directly coupled to a prognostic pressure equation. The projection method of (Chorin, 1968) is used to split the solution procedure in two steps. The first step integrates all the momentum tendencies, except the stated pressure gradient term, explicitly in time to obtain a predicted velocity $\tilde{\boldsymbol{u}}$:

$$\tilde{\boldsymbol{u}} = \boldsymbol{u}_{t_0} + (\partial_t \boldsymbol{u})_{ex}|_{t_0} \Delta t \tag{26}$$

The final velocity after one integration step $\boldsymbol{u}_{t_1}$ has to fullfill the continuity equation:

$$\nabla \cdot \boldsymbol{u}_{t_1} = 0. \tag{27}$$

The not yet known corrective tendency is the neglected pressure gradient $\partial_t \tilde{\boldsymbol{u}} = -\frac{1}{\rho_{ref}} \nabla p|_{t_1}$. Applying the divergence operator on both sides gives:

$$\nabla \cdot \boldsymbol{u}_{t_1} = \nabla \cdot \tilde{\boldsymbol{u}} - \frac{\Delta t}{\rho_{ref}} \triangle p|_{t_1}. \tag{28}$$

Since, as mentioned, the left side has to be zero, one obtains the well-known Poisson equation for pressure, which is to be solved algebraically:

$$\frac{\rho_{ref}}{\Delta t} \nabla \cdot \tilde{\boldsymbol{u}} = \triangle p|_{t_1} \tag{29}$$

The final state is now composed of the fractional tendencies:

$$\boldsymbol{u}_{t_1} = \boldsymbol{u}_{t_0} + \Delta t \left[ (\partial_t \boldsymbol{u})_{ex}|_{t_0} - \frac{1}{\rho_{ref}} \nabla p|_{t_1} \right] \tag{30}$$

Equation 30 is also an example of a first-order accurate in time Euler method. Higher-order multistage Runge-Kutta (RK) methods have the advantages of increased temporal accuracy and larger time stepping. The numerical stability of the integration in most practical examples is constrained by the advective and pressure-gradient tendency. Therefore, all the remaining terms are considered as the minor tendencies, and are evaluated only at every first stage. Different RK methods were tried in our

model framework. It was found that the $2^{nd}$-order midpoint rule (MP-RK2) and the strong-stability preserving $3^{rd}$-order scheme (SSP-RK3) performed reasonably well. Using Taylor expansion, Karam et al. (2019) showed that RK2 and RK3 schemes can preserve their order of accuracy in spite of using just one pressure solve at the final stage. For all the intermediate projection steps, $1^{st}$- and $2^{nd}$-order accurate pressure estimates are interpolated from values at previous time steps. In practical examples, the computational savings were partially sacrificed by a decreased stability of the schemes, requiring smaller time

steps. We found that MP-RK2 maintains stability up to a Courant number of $C = 0.3$, and the SSP-RK3 scheme up to $C = 0.7$ with one final pressure solve. Based on this, SSP-RK3 will be used for all of the applications in this paper and in future.





### 2.4.1 Pressure solution

The discretization of the Laplace operator $\mathbf{P}$ in the Poisson equation (Eq. 29) is obtained by the product of the divergence matrix $\mathbf{D}$ and gradient matrix $\mathbf{G}$ defined by the discrete forms. $\mathbf{D}$ is defined by the flux-balance in Eq. 15. Combining the operators to
the pressure equation, the following sparse linear system is obtained:

$$\mathbf{P}p = \frac{1}{\Delta t}\mathbf{D}\tilde{u} =: b \tag{31}$$

$\tilde{u}$, $p$ and $b$ are the one-dimensional expanded arrays of the corresponding structured fields. Eq. 31 is solved with a geometric multigrid method in parallel using domain-decomposition in two dimensions. The multigrid algorithm consists of applying a smoothing method of choice which is accelerated by coarse-grid corrections. Therefore, a hierarchy of coarse grids is employed
(Brandt and Livne, 2011). 3-d coarsening of the grid is carried out by agglomeration of 8 grid cells to form a coarse grid cell of the next-level grid. For odd grid sizes, a plane of grid-cells with respective orientation is left uncoarsened. This particular multigrid used in combination with finite volume discretizations is often referred to cell-centred multigrid in literature (Mohr and Wienands, 2004).

Particular challenges to the multigrid algorithm include grid stretching and non-smoothly varying coefficients associated with
the diffusive boundaries, both resulting in coefficient anisotropy. An odd grid size results in coefficient anisotropy of the coarse grid operators. Furthermore, Neumann boundary conditions result in a reduced smoothing efficiency near boundaries. In such cases, plane smoothers are much more adequate than their point-wise pendants (Llorente and Melson, 2000). Nevertheless, most of the difficulties were overcome by applying less elaborated methods in the current model.

Based on smoothing analysis, Larsson et al. (2005) give a condition for the optimal location of an uncoarsened plane in the
case of odd grid sizes. For an odd grid size in any direction, the Galerkin coarse-grid approximation (GCA) is much more robust than re-discretization of the Poisson equation. In GCA, the coarse-grid operator is formed algebraically by applying interpolation operators from the left (restriction) and right (prolongation) on the original matrix. The original stencil size of 7 points (without using terrain-following coordinates) is only preserved for piece-wise constant interpolation for both restriction and prolongation. In this case, it is necessary to multiply with a factor of $1/2$ to compensate for the poor approximation quality
(Braess, 1995). Trilinear prolongation would give an adequate approximation quality for constant-coefficient operators, but it failed in our case of highly discontinuous media. As a compromise, the trilinear interpolation operator can be modified to approximate piece-wise constant interpolation near diffusive boundaries. As a drawback of trilinear interpolation, the coarse-grid stencil encompasses 27 points which increases numerical costs of coarse-grid corrections.

The smoothing method employed on each grid has to be efficient for moderate coefficient anisotropies. Yavneh (1996) found
that successive over-relaxation (SOR) is generally superior to Gauss Seidel smoothing (even for isotropic coefficients), and he also derived approximately optimal over-relaxation factors for SOR with red-black ordering applied in multigrid for the solution of anisotropic elliptic equations. An advantage of the red-black ordering is the complete vectorization and parallelization of the algorithm, while a disadvantage is the not optimal cache efficiency, as the computation grid is accessed twice in each iteration (Di et al., 2009). A more cache efficient SOR method provides the standard lexicographic ordering, which however

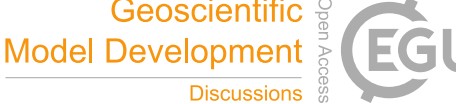

suffers from a degradation toward the less efficient Jacobi relaxation at subdomain boundaries in the parallel implementation. Suitable alternatives for anisotropic problems to SOR are sparse-approximate inverse (SPAI) matrices as smoothers (Tang and Wan, 2000; Bröker and Grote, 2001; Sedlacek, 2012). Depending on the approximation quality of SPAI, which can be controlled by the sparsity pattern and consequently the number of allowed non-zeros, the efficacy of the smoother can be flexibly enhanced in order to tackle the difficult Poisson problem with varying coefficients. Since the Poisson matrix is not time depen-

dent, it pays off to invest computational resources in a good approximation, as it is only necessary once at the beginning of the simulation.

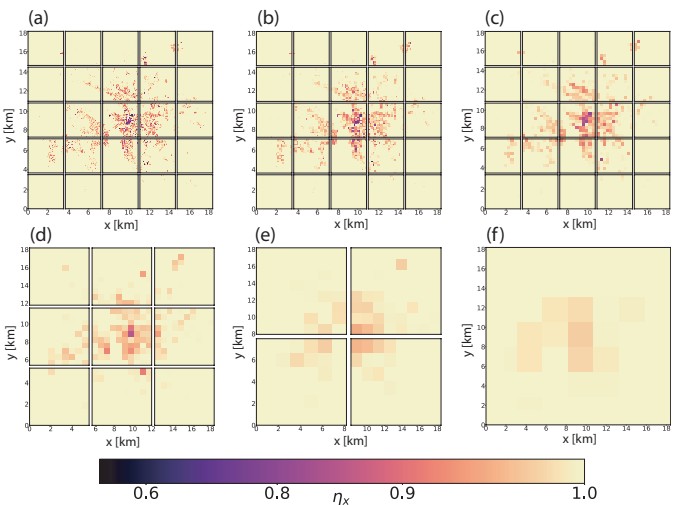

**Figure 2.** Example of a horizontal multigrid domain decomposition, involving 6 grid levels. Depicted are the area-scaling factors of yz-faces for the city of Leipzig at different resolutions, which are used to discretize the Poisson equation. On the coarsest grid, a single processor is left for the computation.

### 2.4.2 Lateral boundary conditions

Before each pressure correction step, global boundary conditions for the next time step are already updated for the intermediate velocity field $\tilde{\boldsymbol{u}}$. This approach naturally leads to a homogeneous-zero Neumann boundary condition for the pressure, which is shown in the following. We require that the updated velocity field satisfies global mass conservation:

$$\oint_{\partial V} \frac{\rho_{ref}}{\Delta t} \tilde{\boldsymbol{u}} \cdot \boldsymbol{n} \, \mathrm{d}a = 0 \tag{32}$$





Using Eq. 32 as the source term of the pressure equation (Eq. 29) and applying Stokes theorem, one arrives at a Neumann
condition for the pressure:

$$
\oint_{\partial V} \frac{\rho_{ref}}{\Delta t} \tilde{\boldsymbol{u}} \cdot \boldsymbol{n} \, \mathrm{d}a = \int_{V} \frac{\rho_{ref}}{\Delta t} \nabla \cdot \tilde{\boldsymbol{u}} \, \mathrm{d}V = \int_{V} \nabla \cdot \nabla p \, \mathrm{d}V = \oint_{\partial V} \nabla p \cdot \boldsymbol{n} \, \mathrm{d}a = 0 \tag{33}
$$

Here $\boldsymbol{n}$ is the unit normal vector on the boundary surface. Since the tendencies of the already updated boundary values are
zero, no projection is applied, which corresponds to the homogeneous-zero condition:

$$
0 = \rho_{ref} \frac{\partial \tilde{\boldsymbol{u}}}{\partial t} \cdot \boldsymbol{n} = \nabla p \cdot \boldsymbol{n} \tag{34}
$$

Note that by already updating the boundary condition for $\tilde{\boldsymbol{u}}$, the pressure boundary condition was homogenized by attributing
the in-homogeneity to the source term $\nabla \cdot \tilde{\boldsymbol{u}}$. An important remark to Eq. 34 is concerning the special case, when terrain-
following coordinates are used. By requiring $\partial_t \omega = 0$ for the contra-variant vertical velocity, the pressure boundary condition
for the top- and bottom-domain boundaries is given by:

$$
0 = \partial_z p - \partial_x h \partial_x p - \partial_y h \partial_y p \tag{35}
$$

In this most general case, the projection still allows variations in the velocity, and it requires the solution of an additional
linear equation (or an additional grid plane in the pressure equation). In order to keep the boundary velocity vector fixed and
to prevent numerical expenses, it is used the trivial condition at the top and bottom boundaries:

$$
\partial_z p = 0 \tag{36}
$$

$$
\partial_x p = \begin{cases} 0 & \partial_x h \neq 0 \\ \text{not specified} & \text{else} \end{cases}
$$

$$
\partial_y p = \begin{cases} 0 & \partial_y h \neq 0 \\ \text{not specified} & \text{else} \end{cases} \tag{37}
$$

In the discrete gradient operator, homogeneous Neumann boundary conditions are implemented by setting all coefficients
associated with the node where the condition applies to zero.

The not yet specified boundary condition for the velocity field has to be compatible with a dynamic mesoscale forcing and
also with the integrability condition of Eq. 32. The workflow is therefore to first dynamically determine inflow and outflow
regions, impose separate appropriate boundary conditions, and then add a correction flux to the outflow regions to balance




the total inflow-region flux. The outflow regions are determined by $C_\perp > 0$, where $C_\perp$ is a normalized convective transport speed out of the domain. $C_\perp$ is evaluated at interior grid points and time step $t$. For simplicity, it is taken $C_\perp = \frac{\Delta t}{\Delta x} \boldsymbol{u} \cdot \boldsymbol{n}$, however more elaborate formulations exist. The transport velocity is further bounded to $0 \le C_\perp \le 1$. This ensures numerical stability of the prognostic radiation equation applied to outflow ghost cells. As proposed by Miller and Thorpe (1981), first order upwinding is used:

$$\boldsymbol{u}_{l+1}^{t+1} = \boldsymbol{u}_{l+1}^t - C_\perp \left( \boldsymbol{u}_{l+1}^t - \boldsymbol{u}_l^t \right) \tag{38}$$

The order of the spatial indexing $l$ is in normal direction to the boundary and not to confuse with the standard interior indexing. The index $l+1$ corresponds to the first ghost cell. Figure 3 shows that the outflow boundary condition with the proposed convective transport speed is well suited to our incompressible model even for highly unsteady flows, like the depicted vortex street in the wake of a cylinder. Individual vortices are not visibly reflected at the boundary, and also in the temporal mean, based on Fig. 3b, the influence of the boundary is not discernible.

Instead of the flexible inflow-outflow condition, a Rayleigh damping layer can be used as another prognostic tendency near the domain top:

$$\partial_t^{dmp} q = -\frac{R(z)}{\tau}(q - q_0). \tag{39}$$

Eq. 39 can be applied to gradually relax any prognostic variable to a prescribed horizontal mean state at the top boundary. $R$ is a ramp function with values between $[0,1]$, $\tau$ the damping time scale and $q_0$ the prescribed boundary value.

The final step is to impose global mass conservation, for which a correction flux $\dot{m}^d$ is integrated for each dimension $d$ separately:

$$\int_{A_{DIR}^d} \rho_{ref} \boldsymbol{u}_{l+1} \cdot \boldsymbol{n} \, da + \int_{A_{RAD}^d} \rho_{ref} \boldsymbol{u}_{l+1} \cdot \boldsymbol{n} \, da = -\dot{m}^d, \; d \in \{x,y,z\}, \tag{40}$$

where $A_{DIR}$ is the surface area on which a Dirichlet condition is imposed, and $A_{RAD}$ the remaining surface area with the radiation condition imposed. It is further $\sum_d A_{DIR}^d + A_{RAD}^d = \partial V$ the total domain-bounding area. The correction fluxes are converted to a corresponding correction speed $u_{\dot{m}}^d$, which is added to the boundary-perpendicular velocity component on $A_{RAD}$. The mass flux is distributed over the radiation-condition area by using for example $C_\perp$ in a weighting function:

$$u_{\dot{m}}^d = \frac{\dot{m}^d \operatorname{sign}(a)}{\rho_{ref}} \frac{C_\perp}{\int_{A_{RAD}^d} C_\perp \, da} \tag{41}$$





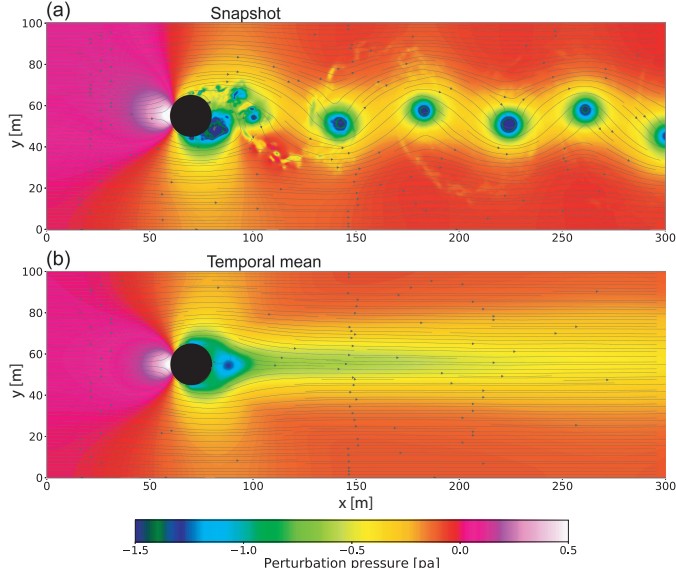

**Figure 3.** LES of 3-D flow past a cylinder. The horizontal grid spacing is uniformly $0.5\,\mathrm{m}$. The cylinder has a diameter of $40\,\mathrm{m}$. The approaching flow is laminar with $u = 1\,\mathrm{m\,s^{-1}}$. Flexible Dirichlet-radiation conditions are imposed on all horizontal domain-boundaries, while periodic boundary conditions are used in $z$-direction. The contour plot depicts the pressure and the streamlines the horizontal velocity field. (a) shows a frame at the instance during a sharply defined vortex is about to cross the right boundary, (b) shows the temporal mean over a representative simulation period.

Attention must be paid to the sign of the correction velocity, as Eq. 41 is multiplied by a negative sign for right-hand sided boundaries. When using terrain-following coordinates, it is the contra-variant vertical flux perpendicular to the bounding sur-

face and used in Eq. 40. The correction speed is simply added to the covariant component $w$.

The flexible boundary condition can principally be applied to any other scalar quantity, however, specifying Dirichlet conditions for advected scalars was found to be sufficient.

## 2.5 Physical processes

### 2.5.1 Subgrid model

For numerical simplicity and efficiency, a static Smagorinsky subgrid model is used (Deardorff, 1970). Since the atmospheric model is operated in the limit of infinite Reynolds numbers, the principal purpose of the subgrid model is to stabilize the numerical simulation through an additional amount of energy dissipation at the shortest wavelengths. Generally speaking, the magnitude of the subgrid fluxes shall be only a small fraction of those of the resolved fluxes, so that any potential non-physical

assumptions in the subgrid model have little influence on the resolved fields. This is a prerequisite for large eddy simulations,





and is one of the main reasons, why the static Smagorinsky model, despite its obvious short comings, is still a popular choice.

The Smagorinsky model relates the subgrid turbulent fluxes to the resolved rates of strain $s_{x,y}$:

$$s_{x,y} = \frac{1}{2} \left( \partial_y u + \partial_x v \right) \tag{42}$$

The subscripts $x$, $y$ here refer to the spatial components.

$$u'_x u'_y = 2\epsilon_k s_{x,y} \tag{43}$$

The terms in Eq. 42 and Eq. 43 are cell averaged values, and overbars are neglected for convenience. The often additionally mentioned anisotropic residual-stress tensor is ignored in the given incompressible case. $\epsilon_{x,y}$ is the eddy viscosity, which for numerical efficiency, is also diagnosed from the rates of strain instead of solving an additional prognostic equation for the

turbulent kinetic energy. $\epsilon_{x,y}$ is denoted in tensor form, as an anisotropic mixing length $l_{x,y}$ is used to reflect grid anisotropy:

$$\epsilon_{x,y} = l_{x,y}^2 |\mathbf{S}| f = (c_s \Delta_{x,y})^2 |\mathbf{S}| f_s \tag{44}$$

$|\mathbf{S}|$ is the Frobenius norm of the stress-rate tensor. $c_s$ is the Smagorinsky constant, which is fixed in a static model. Tests with a boundary layer simulation revealed that in combination with the $5^{th}$-order upwind discretization values of $0.1 < c_s < 0.15$ give good results. Finally, the anisotropy enters via $\Delta_{x,y}$ which is related to the grid spacing and modified by the mean distance

to walls:

$$\Delta_{x,y} = \min \left( \sqrt{\Delta_x \Delta_y}, 1.8 h_x, 1.8 h_y \right) \tag{45}$$

$h_x$ is half of the mean distance between walls orientated in the $x$-direction.

The function $f_s$ introduces the influence of the stratification on the eddy viscosity. It is assumed

$$f_s = \begin{cases} 0 & Ri \geq 0.25 \\ \sqrt{1 - 16Ri} & Ri < 0 \\ (1 - 4Ri)^4 & else, \end{cases} \tag{46}$$

with the Richardson number $Ri$

$$Ri = \frac{g \partial_z \Theta_v}{\Theta_v |S|^2}. \tag{47}$$





For scalar diffusion the eddy viscosity is divided by the turbulent Prandtl number, which is assumed to be $Pr = 2/3$ here.

If not mentioned otherwise, the appearing spatial derivatives are discretized with second-order differences. To obtain the strain rate components and the eddy viscosity on different stencil points (cell faces areas and cell centers), linear interpolation is used. The subscale tendency is approximated by the divergence of diffusive fluxes. To account for the definition of the velocity components on the cell faces, shifted grids are introduced. For example, diffusion of the $u$-component requires a grid shifted by $\Delta_x/2$, for which the scaling fields are obtained by linear interpolation.

The diffusive fluxes are given by:

$$f_{x,y} = 2\epsilon_{x,y} s_{x,y} \tag{48}$$

For diffusion of the $u$-component, the fluxes in the 3 spatial directions are $f_{x,x}$, $f_{x,y}$ and $f_{x,z}$. Those of the other components are obtained by permuting the subscripts. For a scalar quantity $q$, the required fluxes are $f_{x,x}$, $f_{y,y}$ and $f_{z,z}$, wherein the rates of strain are replaced by the gradient components.

### 435   2.5.2   Surface fluxes

Surface fluxes for momentum, heat and moisture are parameterized, using Monin-Obhukov similariry theory (Louis, 1979). The vertical transfer coefficient $C_z$ is given by transforming the logarithmic-wind law:

$$C_z = \frac{k^2}{\log^2\left(z/z_0\right)} \tag{49}$$

$k = 0.4$ is the Von Kármán constant, $z_0$ the surface roughness length, and $z$ the height difference from the modeled surface
to the grid level where the parameterization is evaluated. In order to alleviate the often encountered problem with the log-layer mismatch (Yang et al., 2017), it is advantageous to take the second or third grid level above respective surface, since close to the surface turbulence is not adequately resolved.

The momentum sinks from horizontal surfaces are:

$$\frac{\partial u'w'}{\partial z} = -\frac{A_z}{\Delta V} f_m C_z u \sqrt{u^2 + v^2} \tag{50}$$

$$\frac{\partial v'w'}{\partial z} = -\frac{A_z}{\Delta V} f_m C_z v \sqrt{u^2 + v^2} \tag{51}$$

Analogously, the source terms for heat and moisture from horizontal surfaces are:

$$\frac{\partial \Theta'w'}{\partial z} = -\frac{A_z}{\Delta V} f_h C_z \left(\Theta - \Theta^s\right) \sqrt{u^2 + v^2} \tag{52}$$



$$\frac{\partial Q_v' w'}{\partial z} = -\frac{A_z}{\Delta V} f_h C_z \left(Q_v - Q_v^s\right) \sqrt{u^2 + v^2} \tag{53}$$

$\Theta^s$ is the surface potential temperature, and $Q_v^s$ the surface specific humidity. $A_z$ is the total exposed horizontal surface within the grid cell, and $\Delta V$ the effective cell volume. $f_m$ and $f_h$ are stability functions, and $z_0$ the surface roughness length. $f_m$ is based on the Bulk Richardson number, which is the fraction of buoyant to shear energy production and calculated as:

$$RiB = \frac{g\left(\Theta - \Theta^s\right) z}{\Theta^s \left(u^2 + v^2\right)} \tag{54}$$

In stable conditions, $RiB > 0$, and the stability functions are empirically determined by Doms (2011) for land surfaces:

$$f_m = \frac{1}{1 + 10 RiB \left(1 + 5 RiB\right)^{-0.5}} \tag{55}$$

$$f_h = \frac{1}{1 + 15 RiB \left(1 + 5 RiB\right)^{0.5}} \tag{56}$$

Conversely, in unstable conditions $RiB < 0$, and

$$f_m = 1 + \frac{10|RiB|}{1 + 75 C_z \left[\left(\frac{z}{z_0}\right)^{0.33} - 1\right]^{1.5} \sqrt{|RiB|}} \tag{57}$$

$$f_h = 1 + \frac{15|RiB|}{1 + 75 C_z \left[\left(\frac{z}{z_0}\right)^{0.33} - 1\right]^{1.5} \sqrt{|RiB|}} \tag{58}$$

Sources and sinks from vertical building walls are treated similarly, with the exception that the stability function is set to unity. For $x$-orientated surfaces, $z$ is replaced by half of the average distance between surfaces in the equation for the transfer coefficient. Analogously, $A_z$ is replaced by $A_x$ for the total projected surface area with $x$-orientation.

    The surface fields $\Theta^s$ and $Q_v^s$ can be considered as part of the external forcing and have to be provided either by the hosting
mesoscale model or field-interpolated measurements.

### 2.5.3 Turbulence recycling scheme

Wu (2017) gives an overview of various turbulence generation methods to provide turbulent inflow conditions. Among the different methods, a turbulence recycling scheme was implemented in the model, as it is computationally efficient and can be





applied to a wide range of different domains and flow types. One peculiarity of the used scheme is that the recycling plane can
be placed at an arbitrary distance to the inflow boundary within the computation domain, and all 4 horizontal boundaries are
considered as potential inflow boundaries, unless periodic boundary conditions are specified. The only requirement is that the
plane distance is at least several integral length scales to prevent a spurious periodicity of the recycled turbulent features. In
some cases, it may even practical to place the recycling planes near the respective opposite boundaries, if the inflow conditions
shall be those from an urban boundary layer. In this case, no extra development fetch is needed.

At each model time step, a horizontal filter is applied on the velocity components within the recycling plane. By setting
the characteristic filter width $w_r$ to the integral length scale, the mesoscale variations are spared, which is important in case
of spatially and temporally varying boundary conditions. Vertical filtering is not feasible due to the strong vertical wind shear
within the boundary layer. The filtered velocity component is subtracted to obtain the small-scale fluctuation component:

$$u(z,y)' = u(z,y) - <u(z,y)>_{w_r} \tag{59}$$

Equation 59 assumes a boundary perpendicular to the flow in x-direction. Next, the turbulent intensity is re-scaled to the
target value, after which the fluctuation field can be added to the inflow boundary field $u_{in}$ of the large scale flow $u_{ls}$.

$$u_{in}(z,y) = u_{ls}(z,y) + min\left[a_{max}, \frac{||u'_{tar}||_2(z)}{||u'||_2(z)}\right]u'(z,y) \tag{60}$$

$a_{max}$ is used to limit the artificial amplification of turbulence shortly after model initialization.

The filtering operation, as well as the calculation of the turbulent intensities require communications in the parallel im-
plementation. In order to keep a maximum degree of parallelism in the model, decentralized filtering is used, instead of the
much simpler method of transferring all the data to a root node. Before filtering, additional ghost cells are exchanged, whose
communication effort can be substantial for large filter widths. In this regard it is advantageous and adequate to use an efficient
box filter. A final communication is potentially necessary to transfer the data to the inflow boundary, if the recycling plane is
located on another subdomain. Overall, despite the communication overload, the computational costs of the recycling scheme
were found to be only 1% to 2% of total costs on average.

## 3 Model evaluation

A series of simulations of different complexity is carried out in order to assess the model accuracy and demonstrate the
capabilities of the numerical core. The first model experiment is an advection test taken from Calhoun and LeVeque (2000).
A tracer is advected within a rotating annulus. Due to to the large number of intersected grid cells in manyfold ways, this
test can be considered as challenging for the empirically determined order of convergence. In the second test, which is also
reported in Calhoun and LeVeque (2000), a wave of a test tracer is advected through an irregular obstacle field. The stationary


wind field is a numerically-approximated potential-flow solution obtained with a single projection step. A sensitivity study is performed to determine the grid-spacing sensitivity of the advection routine in combination with the projection method. The

third basic example to evaluate the dynamic core is the rising bubble simulation of Wicker and Skamarock (1998). While the benchmark simulation is compressible, it will be determined how well the anelastic approximation in this model compares to it. The actual evaluation part is completed with the simulation of the wind-tunnel experiment "Michelstadt" (Berbekar et al., 2013). It provides a high-quality standard dataset for LES-type dispersion model evaluation. As a last example, which is orientated towards a more practical application, the models capability is demonstrated to simulate meteorology and air-

pollution dispersion under non-neutral stratification over non-uniform terrain.

### 3.1   Annulus advection test

In the advection test described in Calhoun and LeVeque (2000), a circumferential flow field inside an annulus with the inner radius $R_1 = 0.75$ and outer radius $R_2 = 1.25$ is defined by the following potential-flow function:

$$\Psi = \begin{cases} -\frac{\pi}{5}r^2 & R_1 < r < R_2 \\ -\frac{\pi}{5}R_1^2 & r \leq R_1 \\ -\frac{\pi}{5}R_2^2 & r \geq R_2. \end{cases} \tag{61}$$

$r = \sqrt{x^2 + y^2}$, and $(x, y)$ are the coordinates of the grid-corner points. By differencing $\Psi$, the velocity components on the grid edges are obtained:

$$u = \frac{\Psi(i+1, j) - \Psi(i, j)}{\eta_x \Delta y}$$
$$v = -\frac{\Psi(i, j+1) - \Psi(i, j)}{\eta_y \Delta x} \tag{62}$$

Analogous to the 3-D case, $\eta_x \Delta y$ and $\eta_y \Delta x$ are the effective lengths of the cell edges normal in $x$-direction and $y$-direction,

respectively. It can be easily shown that the resulting velocity field satisfies $\nabla \cdot \boldsymbol{u} = 0$. The initial concentration of the advected tracer is given by

$$c_0 = \frac{1}{2}\left\{\left[5(\phi - \frac{\pi}{3})\right] + \left[5(\frac{2\pi}{3} - \phi)\right]\right\}. \tag{63}$$

$\phi$ is the azimuth defined in a mathematically positive sense. One revolution of the tracer takes $5\,\mathrm{s}$ of simulation time, after which the analytical solution is identical to the initial state.


The case is simulated with increasing grid sizes of $50 \times 50$, $100 \times 100$, $200 \times 200$, $400 \times 400$ and $800 \times 800$, respectively. The integration step size is halved at each doubling of the resolution, starting at $\mathrm{d}t = 0.016\,\mathrm{s}$.





Figure 4a depicts the relevant hemisphere of the annulus with the solution after one revolution using the $800 \times 800$ grid.
Figures 4b-f show the deviations from the exact solution on the different grids. Not unexpectedly, the largest error magnitudes

occur towards the edges of the annulus, but generally diminish with increasing grid size. The respective error norms are
shown in Table 1. $L_\infty$ is the maximum error magnitude, which convergences with an average rate of $R_\infty = 0.8$. In contrast,
Calhoun and LeVeque (2000) did not observe convergence in the $L_\infty$ norm for the Péclet numer $Pe = \infty$ case. In the $L_1$ norm
(mean absolute differences), the schemes accuracy is closer to second order with an average convergence rate $R_1 = 1.73$. The
convergence rate is not significantly affected by the order of accuracy of the reconstructions, which is $5^{th}$ by default. This

suggests that the flux limiting has a profound impact on the results of this test case.

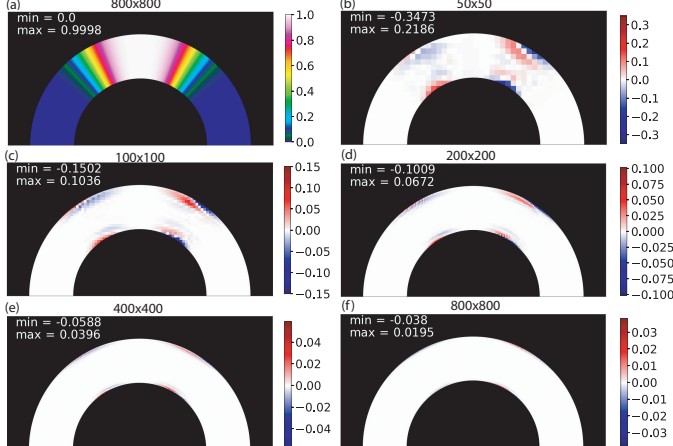

**Figure 4.** Test tracer distribution in arbitrary units. (a): Solution after one revolution for a $800 \times 800$ grid. (b-f): Difference plots of difference between numerical and exact solution of the experimental convergence study for grid sizes $50 \times 50$, $100 \times 100$, $200 \times 200$, $400 \times 400$, and $800 \times 800$.

| | $50 \times 50$ | $100 \times 100$ | $200 \times 200$ | $400 \times 400$ | $800 \times 800$ |
|---|---|---|---|---|---|
| $L_1$ | $3.194 \times 10^{-3}$ | $9.169 \times 10^{-4}$ | $3.079 \times 10^{-4}$ | $8.873 \times 10^{-5}$ | $2.607 \times 10^{-5}$ |
| $L_\infty$ | 0.3473 | 0.1502 | 0.1092 | $5.880 \times 10^{-2}$ | $3.800 \times 10^{-2}$ |
| $R_1$ | – | 1.801 | 1.574 | 1.795 | 1.767 |
| $R_\infty$ | – | 1.209 | 0.574 | 0.779 | 0.630 |

**Table 1.** Error norms $L_1$ and $L_\infty$ and resulting rates of convergence $R_1$ and $R_\infty$ for the different model grids in Fig. 4





## 3.2 Advection trough an obstacle field

This 2-D test initially consists of an approximation of a potential flow solution for an irregular obstacle field. Therefore, one step of the explained projection method is applied on the initial wind field defined by $u = 1$ and $v = 0$. The resulting potential flow field is used to advect a test-tracer front, which is characterized by the left inflow boundary condition:

$$c = \begin{cases} 1 & t \leq 40\,s \\ 0 & t > 40\,s \end{cases} \tag{64}$$

For the transversal-flow direction, periodic boundary conditions are used.

The reference simulation is carried out on a domain with $200 \times 100$ grid cells and a uniform grid spacing of $1\,\text{m}$. The obstacles are circular with radii ranging from $5\,\text{m}$ to $10\,\text{m}$. The simulation is repeated on coarser grids with dimension sizes of $100 \times 50$, $50 \times 25$, and $25 \times 13$, respectively. Figure 5 shows the simulation results at $t = 150\,s$. The obstacles are well resolved on the grid with $1\,\text{m}$ spacing, but become more and more diffuse with decreasing grid resolution towards $8\,\text{m}$ for the coarsest grid. The initially planar wave is delayed and deformed by the obstacles. The qualitative impression is that the shape of the wave is not really sensitive to the grid resolution. Even in the most diffuse case, the position and shape of the wave matches that of the higher resolved simulations well. The wave dispersion can be quantified by considering the washout curves shown in Fig. 6, which are the spatially averaged concentrations at the outflow boundary versus time. For the finest grid, the concentrations at the outflow boundary start to rise after $t = 145\,s$ and peak at about $t = 185\,s$. At $t = 250\,s$ most of the wave is advected out of the domain. Remarkably, as already found by (Calhoun and LeVeque, 2000), the curve is not sensitive to the grid resolution up to $4\,\text{m}$, which is at the transition where obstacles start to become diffuse. At the coarsest resolution of $8\,\text{m}$, the peak is slightly broader, peak concentrations are lower and the peak occurs earlier by about $5\,s$. At this grid resolution, the intrinsic diffusion of the advection scheme becomes important, as the resolution capability is around 6 grid points which is barely enough to resolve the wave.

## 3.3 Rising thermal

In the rising thermal simulation described in Wicker and Skamarock (1998), the Euler equations without diffusion are solved on a 2-D domain with a height of $10\,\text{km}$ and a width of $20\,\text{km}$. The grid spacing is uniformly $125\,\text{m}$. Within the otherwise constant virtual potential temperature field of $\Theta_v = 300\,K$, a circular perturbation is placed:



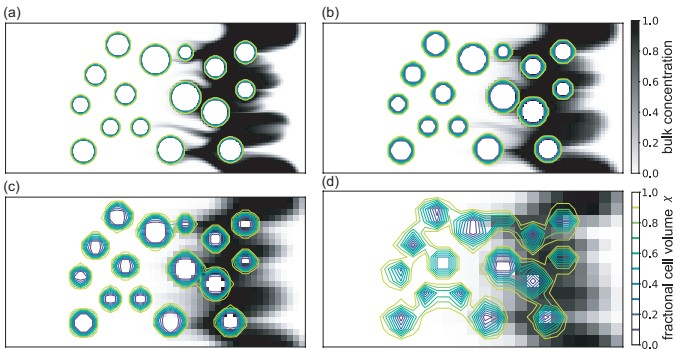

**Figure 5.** Map plots of the concentration field of a test tracer after a simulation time of $150\,\mathrm{s}$. The obstacles are drawn by contours of the volume-scaling field $\chi$. Shown are the results of the flow simulations for the grids (a) $200 \times 100$ cells, (b) $100 \times 50$ cells, (c) $50 \times 25$ cells, and $25 \times 13$ cells.

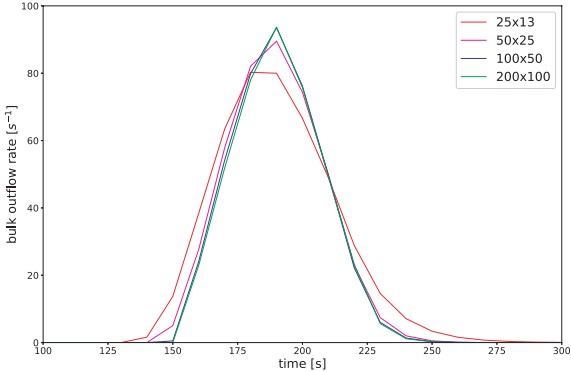

**Figure 6.** Washout curves of the test tracer for the different grid resolutions in Fig. 5.

$$\Delta\Theta_v = \begin{cases} 2\cos^2\left(\frac{\pi r}{2L}\right) & r \leq L \\ 0 & r > L, \end{cases}$$
$$r = \sqrt{x^2 + (z - 2\,\mathrm{km})^2},$$
$$L = 2\,\mathrm{km} \tag{65}$$

The initial vertical velocity is $w = 0$ and the horizontal velocity $u = 20\,\mathrm{m\,s^{-1}}$. Periodic lateral boundary conditions are used
and a rigid boundary is placed at the domain top. Due to buoyant forces, the thermal starts rising, while it is constantly advected



to the right and eventually re-enters the domain at the left boundary. Finally, at the simulation time of $t = 1000\,\mathrm{s}$, the thermal is again situated in the center of the domain.

Figure 7a shows the evolution of the thermal based on contours of $\Theta_v$ at the time steps $t = 0\,s$, $t = 350\,s$, $t = 650\,s$ and $t = 1000\,s$. Two distinct, symmetric rotors develop, and at the time $t = 1000\,s$, the overall appearance of the thermal matches

that of the original simulation by Wicker and Skamarock (1998). However, in our simulation the thermal is more compact, as it is confined between $x = \pm2300\,\mathrm{m}$ and the peak height is at about $8100\,\mathrm{m}$. In the original simulation it is confined between about $x = \pm2600\,\mathrm{m}$ and below $8500\,\mathrm{m}$. Whether this slight discrepancy stems from the Boussinesq approximation or the different numerical schemes used, can not be finally clarified. By scale analysis, the magnitude of the buoyant acceleration is about one or two orders less than that of the inertial acceleration in the given example, so the Boussinesq approximation may

indeed apply well here. The $5^{th}$-order upwind scheme used here is much less diffusive than the $3^{rd}$-order scheme used by Wicker and Skamarock (1998). On the other hand, the flux limiting introduces more diffusion at sharp gradients and ensures a positive solution ($\theta_v \geq 300\,\mathrm{K}$). This can be an explanation for the smoother contour lines inside the rotor. A slight asymmetry from the lateral advection can be noticed, most evident in the contours of vertical wind speed in Figure 7b. This asymmetry can be slightly reduced by decreasing the integration step size (not shown). The combination of the advection scheme with the

$3^{rd}$-order SSP-RK3 time scheme gives stable results up to a Courant number of $C = 0.7$. Positivity of the solution is preserved up to $C = 0.5$.

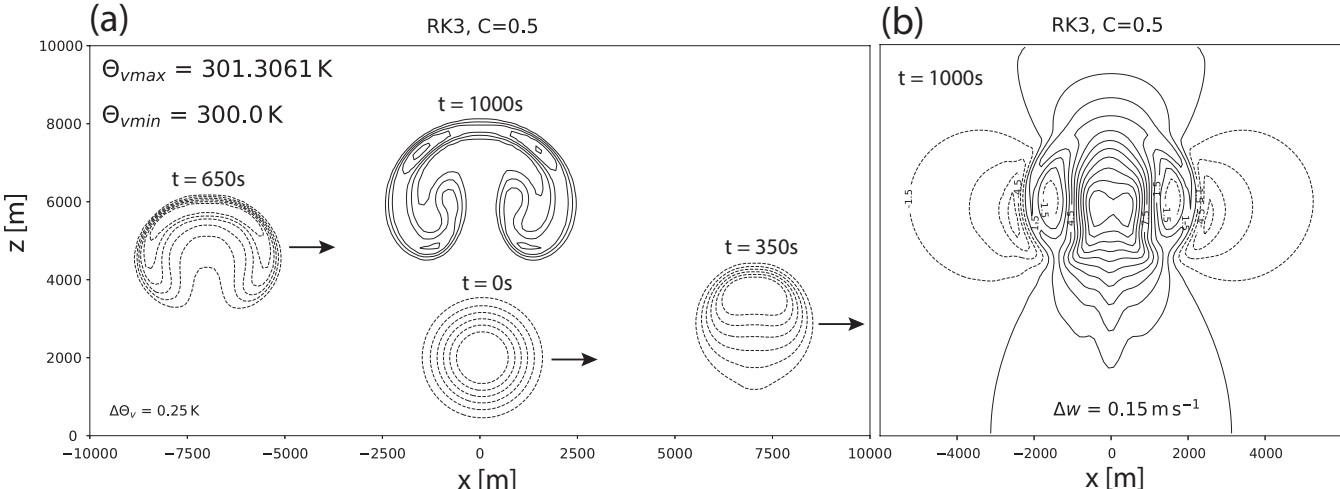

**Figure 7.** Rising thermal simulation with lateral advection: (a) Contours of virtual potential temperature at different time steps. The initial and intermediate states are drawn with dashed lines, the final state at $t = 1000\,s$ with solid lines. The contours with $\theta_v = 300\,\mathrm{K}$ are omitted. (b) Contours of vertical wind speed at $t = 1000\,s$. Dashed lines are used for negative values.



### 3.4 Michelstadt wind-tunnel experiment

The wind tunnel experiment "Michelstadt", which was carried out in the wind tunnel WOTAN (Lee et al., 2009) of the University of Hamburg, is used to evaluate the model accuracy and reliability against a physically based dataset. The benefits of using
wind tunnel data for numerical model evaluation are the accurately controlled and determined approaching-flow conditions, the high spatial and temporal resolution of measurements, and the high statistical significance of the data compared to field data (Schatzmann et al., 2017). Assuming the validity of the scale invariance, wind-tunnel experiments provide a reliable basis of comparison under idealized conditions (e.g. neutral stratification, absent surface fluxes, temporally constant and horizontally homogeneous approach-flow statistics).

"Michelstadt" is the name of a fictitious model city district, designed at an assumed scale of 1:225. The district spans an area of roughly $1350\,\text{m} \times 850\,\text{m}$ at the full scale. The simplified geometry is based on a typical Central European downtown area, with spacious polygonal courtyards surrounded by residential building units. The orientation and length of street-canyon sections are highly variable, and isolated squares are present. All roofs are approximated by a flat surface (flat-roof model) and their full-scale heights range from $15\,\text{m}$ to $25\,\text{m}$. See Fig. 8a for an overview of the building layout.

The approaching wind field is characterized by a horizontally homogeneous boundary-layer flow with neutral stratification. The parametric surface-roughness length is comparatively large with approximately $1.4\,\text{m}$, implying the seamless incorporation of the test section into a more extensive urban area. Turbulence of the approaching flow is initiated by an arrangement of vertical spikes, and the passage of the flow over an extended pattern of surface-roughness elements establishes turbulent kinetic equilibrium in the wind-tunnel experiment. In order to characterize the statistical properties of the approaching flow
more accurately, experiments were carried out where the entire wind-tunnel floor was covered with roughness elements. The actual experiments with the model city were carried out using two different wind directions of the approach flow ($0\,^\circ$, $180\,^\circ$), where the reverse $180\,^\circ$ direction can be used to verify the robustness of model results after parameter tuning with the reference case $0\,^\circ$. The experimental datasets consist of time resolved flow and dispersion measurements, from which also the stationary statistics were calculated. A dense array of sensors covers horizontal wind measurements within planes at heights $2\,\text{m}$, $9\,\text{m}$,
$18\,\text{m}$, $27\,\text{m}$, and $30\,\text{m}$ over a restricted area. For the dispersion modeling, neutral-buoyant gas was released at different locations on the floor (see Fig. 8 for the locations of the release points). Release points S2 and S4 were used for the approach-flow direction of $0\,^\circ$, S6, S7 and S8 for the reverse direction. S5 was used for both wind directions. Depending on the tested scenario (fast response vs. average air pollution), the releases were in puffs or continuous. The concentrations, converted to dimensionless units of ppmv, were measured using flame-ionization detectors all placed at a full-scale height of $7.5\,\text{m}$. Further information
on the experiment and the datasets are provided in Baumann-Stanzer et al. (2015).

The numerical simulations are performed at the full scale using a series of grids with horizontal resolutions of $5\,\text{m}$, $10\,\text{m}$, $20\,\text{m}$, $40\,\text{m}$, and $80\,\text{m}$. The $5\,\text{m}$ resolution is used as for reference, and the coarser domains are used to test the sensitivity of the dispersion simulation to the spatial resolution. The vertical grid spacing near the surface ranges from $2\,\text{m}$ for the finest grid to $7\,\text{m}$ for the coarsest ones, and is increasingly stretched beyond $30\,\text{m}$ above surface. The effective domain height is approx-
imately $600\,\text{m}$, which corresponds to the scaled wind-tunnel height. Above this height the wind components are dampened to





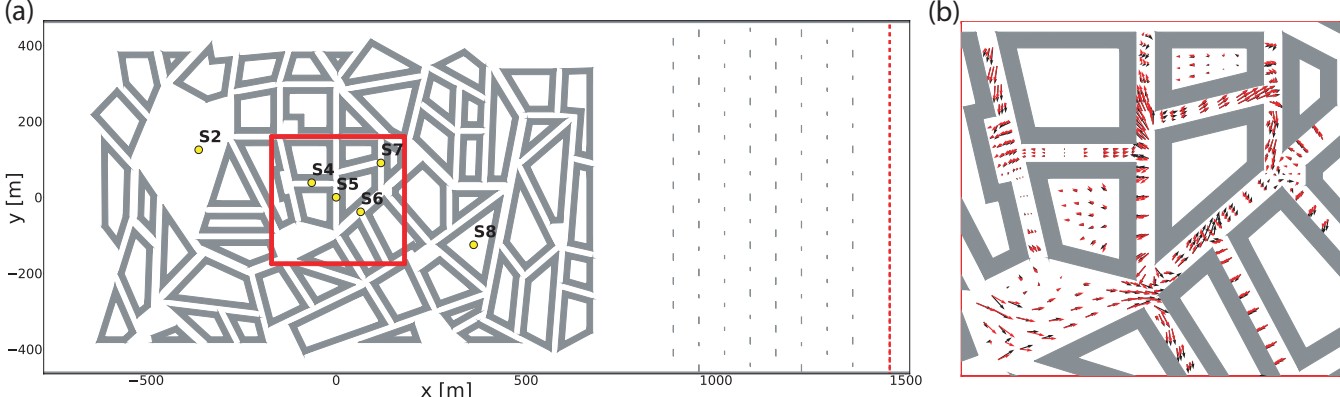

**Figure 8.** (a) Depiction of the model domain for the numerical simulation of the wind tunnel experiment "Michelstadt". Rigid boundaries, including the buildings, the side walls and the roughness elements, are drawn using grey color. The yellow circles mark the tracer-gas release points. The red dotted line is at the position of the turbulence recycling plane. The area within the red bounding box contains the horizontal wind measurements arranged in horizontal planes. (b) Comparison of measured (black arrows) and modeled (red arrows) time-averaged horizontal wind vectors at $2\,\mathrm{m}$ height.

the horizontally and temporally average state.

A first precursor simulation is run with periodic boundary conditions to obtain laminar and turbulent inflow profiles for the full vertical domain extend used to drive the actual experimental simulations containing the buildings and test-tracer release points. In this precursor simulation, which uses a shorter domain length of about only $1\,\mathrm{km}$, buildings are not present and the

entire rigid bottom-domain boundary is covered with roughness elements, with the arrangement adapted from the experiment. In order to model the effect of the lateral wind-tunnel confinement, rigid walls are placed at the flow-perpendicular boundaries. The modeled statistics of the established neutrally-stratified boundary layer flow are re-scaled to the reference wind speed of $6\,\mathrm{m\,s}^{-1}$ at $50\,\mathrm{m}$ height. The obtained horizontally averaged vertical profiles are vertically interpolated for the coarse-grid simulations. In the experimental simulations containing the model city, turbulent approach-flow conditions with the correct

target intensities are generated using the turbulence-recycling scheme. The recycling plane is placed well downstream of the model city and after a short pattern of roughness elements near the outflow boundary (Fig. 8a). This allows the application of a much shorter additional domain fetch for turbulence recycling, as the total recycling fetch can be much larger using the upstream domain. This particular positioning of the recycling plane can be justified by the similar parametric roughness length of the elements and the model city. Apart from that, the extracted fluctuation intensities are always re-scaled to the values of

the target wind field.





### 3.4.1 Inflow profile

Figure 9 shows the modeled and re-scaled horizontally averaged statistics of the boundary-layer flow using the precursor periodic domain. The experimentally obtained profiles are included for comparison. In the modeled mean horizontal wind speed, the roughness layer extends up to $20\,\mathrm{m}$ above ground, and is proceeded by the logarithmic Prandtl layer. The slope of

the modeled wind profile in the semi-logarithmic depiction matches the observed profile well. The modeled turbulent intensities are generally too low by up to 20 % when compared to the observed ones. Partly this can be attributed to the neglected subgrid and numerical diffusion. Apart from that, the peak values at about $400\,\mathrm{m}$ height are probably reflections from the domain top, not fully captured by the dampening. The measurements below a height of $20\,\mathrm{m}$ are more difficult to compare since they are located within the roughness layer, and their values strongly depend on the relative location to the nearby roughness elements.

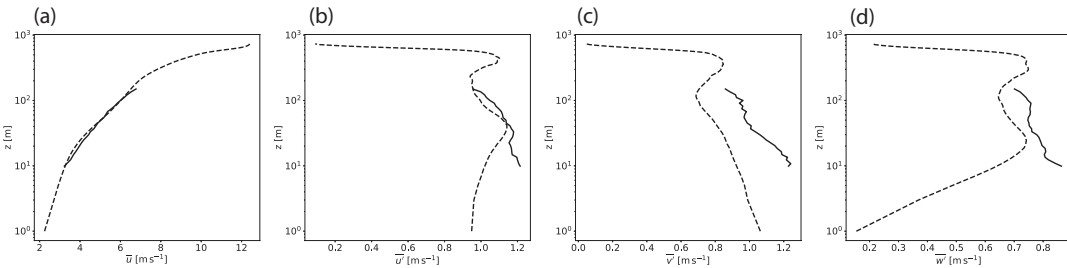

**Figure 9.** Modeled (dotted lines) and measured (full lines) mean and turbulent statistics of the approach flow in the "Michelstadt" wind tunnel experiment. Depicted are (a) the temporal mean velocity component $\overline{u}$, (b-d) the grid-scale turbulent intensities $\overline{u'}$, $\overline{v'}$, and $\overline{w'}$, respectively.

### 3.4.2 Horizontal wind evaluation

The modeled horizontal wind is evaluated on the reference grid with $5\,\mathrm{m}$ horizontal resolution. Figure 8b gives a first qualitative impression of the agreement, as it shows the time-averaged horizontal wind vectors both for the model and the measurements within the plane of $2\,\mathrm{m}$ height. Overall, the model is capable of reproducing the measured flow pattern. However, the modeled wind direction does not always match the corresponding measured vector well, most notably near some intersections. The

scatter plots for wind speed (Fig. 10) and wind direction (Fig. 11) give a more quantitative and conclusive picture. The accuracy of modeled wind speed is high, when taking the normalized mean standard error (NMSE) as a proxy, whose value is consistently below 0.1. The fractional bias (FB) shows only a very slight underestimation of wind speeds (0.02 - 0.07). This is not very surprising, since the model resolution of $5\,\mathrm{m}$ is not high enough to fully resolve the small-scale circulations within the urban canopy.




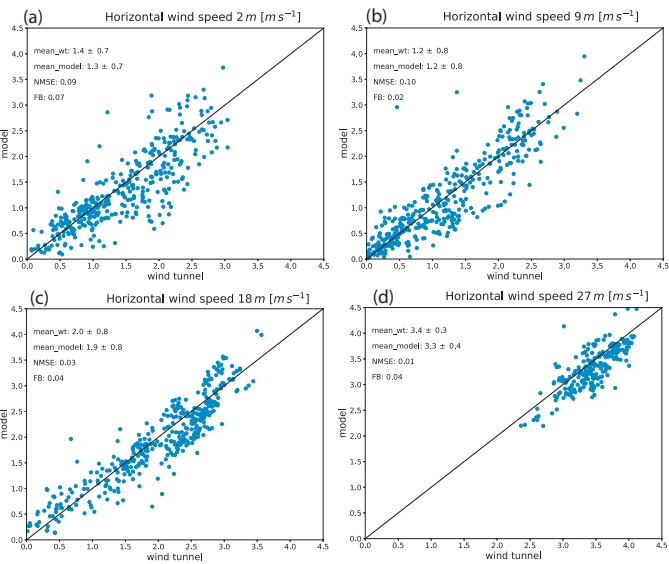

**Figure 10.** Scatter plot of modeled vs. measured horizontal wind speed within planes at different heights. For a quantitative comparison, NMSE and FB are calculated.

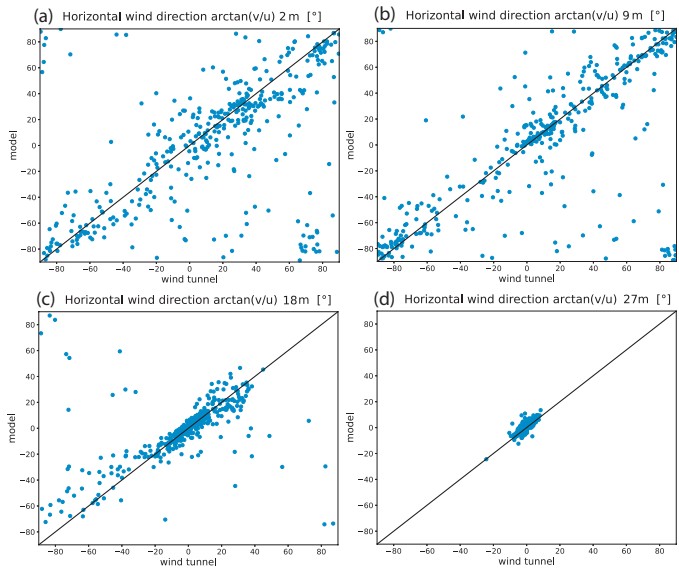

**Figure 11.** Scatter plot of modeled vs. measured horizontal wind direction within planes at different heights.





### 3.4.3 Dispersion simulation evaluation

The dispersion of point-source emissions is evaluated in the continuously emitting mode. The resulting time-averaged modeled concentrations are interpolated to the detector sides and paired with corresponding measurements. As proxy for the quality of model results in comparison to the measurements, the normalized mean standard error (NMSE), the fractional bias (FB) and the fraction of within factor 2 (FAC2) are calculated. Based on the guidelines presented in Baumann-Stanzer et al. (2015), acceptance criteria for a valid simulation are NMSE $< 6$, $|\text{FB}| < 0.67$ and FAC2 $> 0.3$.

The simulations with the approach flow direction of $0\,°$are used to optimize the model configuration with respect to these test criteria, while the reverse flow direction is used to validate the robustness of model results using the same parameter configuration. One important model tuning parameter for example is the Smagorinky constant, which was set to $c_s = 0.15$ for all simulations. However, for the simulations with $40\,\text{m}$ and $80\,\text{m}$ resolution the vertical mixing length was increased to $20\,\text{m}$ within the urban canopy to balance the underrepresented building-induced vertical mixing at such coarse resolutions.

Among the tested sources, S2 is the only one emitting into a quite open area, whereas all other sources are placed within street canyons or courtyards. Thus, S2 is probably the least difficult to simulate. Figure 12 gives an impression of the simulated time-averaged plumes resulting from S2 at the height of detectors. Figure 12a shows the reference simulation with the highest grid resolution. The overall qualitative agreement with measurements seems very good, except for a street canyon in roughly -45 °direction and in close proximity to the source, wherein concentrations are too low. Not surprisingly, increasing the grid spacing results in a deterioration of the qualitative agreement with measurements and with the reference simulation. For example, increasingly less tracer gas is advected inside the flow-parallel street canyon, where most of the detectors reside. Instead of, the plumes become increasingly smeared over a wider area. This is especially evident at the $20\,\text{m}$ grid spacing, whereas at the even coarser resolutions of $40\,\text{m}$ the agreement with measurements improves again. This behaviour is only observed for this particular source. The grid resolution of $80\,\text{m}$ clearly shows the least accurate results. While the dispersion pattern still resembles those of the better resolved simulations and shows the imprint of buildings, concentrations are too high in the down-wind swath. At this resolution, dynamics induced by buildings and important for vertical mixing cannot be properly represented anymore.

For the quantitative evaluation of the presented simulations, the paired data is presented as scatter plots in Fig. 13 and the aforementioned statistical acceptance parameters are derived. For the reference case, most of the model data is tightly distributed near the bisecting line. In fact, the model accuracy is very good (NMSE$= 0.10$), with only a slight positive bias toward too low values (FB$= 0.12$) and only few outliers present (FAC2$= 0.84$). The decrease in model accuracy with increasing grid spacing is evident in the scatter plots, as modeled values tend to be too low for high concentrations measured and vise versa, respectively. This results in a steady increase in NMSE$= 0.25$ for the $10\,\text{m}$ grid spacing and NMSE$= 1.35$ for the $20\,\text{m}$ grid spacing. Since FB is more sensitive to deviations at the upper end of the logarithmic scale, the smearing also results in an increasingly positive bias (FB$= 0.17$ and FB$= 0.65$). The trend is however reversed at the even coarser resolution of $40\,\text{m}$, resulting in an improvement of model results for this particular source. Finally, the $80\,\text{m}$ case shows a large negative bias (FB$= -0.57$) and a value of FAC2$= 0.32$ at the verge of acceptance. Table A1 summarizes the statistical parameters





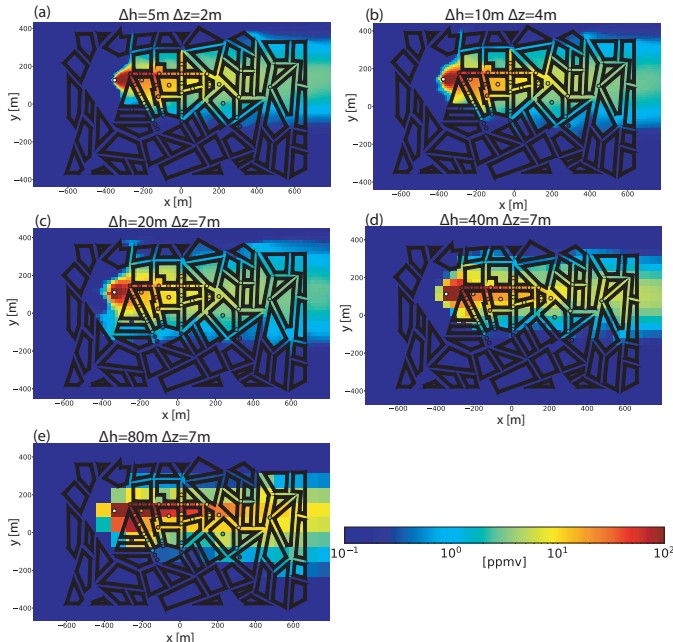

**Figure 12.** Map plots of modeled concentration fields in ppmv at $7.5\,\mathrm{m}$ height for source S2 and different grid resolutions. The black circles mark the location of measurements for this particular source, and the fill color indicates the measured concentration according to the color bar of the map plots.

for all other cases simulated. Based on it, it can be concluded that while moving to ever coarser grid resolutions the quality

of model results generally declines (mostly evident in NMSE value), but not to an extend to compromise model reliability at $40\,\mathrm{m}$ resolution, where buildings are represented diffusely. Only one source located within a courtyard proofed to be more problematic at $40\,\mathrm{m}$ grid spacing, which is due to the difficulty in modeling diagonally orientated building walls as impermeable as they are. Using the $80\,\mathrm{m}$ grid, the model still performs acceptable for some of the sources. Figure 14 shows scatter plots of the data pairs collected from all simulated cases with a given resolution, whose statistical results are again summarized in Tab.

A2. It shows how the reliability of model results is not very sensitive to grid spacing down to $40\,\mathrm{m}$ resolution. For example, FAC2 decreases from a value of $0.8$ at $5\,\mathrm{m}$ to $0.61$ at $40\,\mathrm{m}$ grid spacing. Conversely, NMSE increases from $0.54$ to $2.75$, which is still well within the acceptable range. The average fractional bias is below FB$= 0.2$ for all simulations, except the $80\,\mathrm{m}$ case, where it is much larger (FB$= -0.35$). In this regard, the increased vertical mixing length showed to be an effective tuning option in combination with the $40\,\mathrm{m}$ resolution to keep the bias comparatively low (FB$= -0.17$). It has to be kept in mind that

this test uses isolated point sources. When applied to a more realistic scenario with traffic emissions modeled for example by emission lines, it can be hypothesized that scattering of the data would be of less concern, since the pollution is more widely distributed horizontally. Therefore, ultimately, the most important reliability measure in our view is the FB value, as it is a proxy whether the model will under- or over-estimates air pollution. In this regard, the $80\,\mathrm{m}$ resolution is the least accurate,





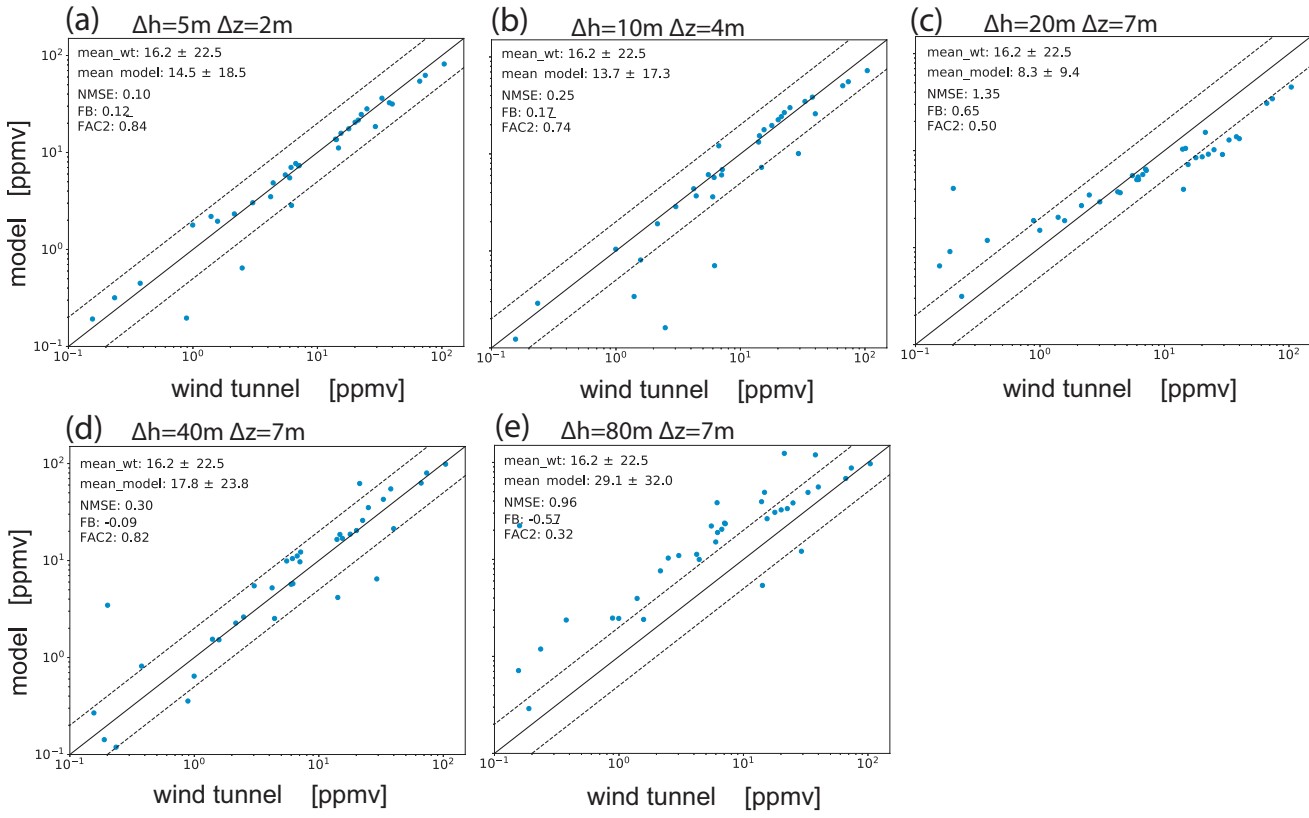

**Figure 13.** Scatter plots of measured vs. modeled concentrations for source S2 and different grid resolutions. The dotted lines confine the region within a factor of two of measurements.

and it is currently not aspired without the use of a more sophisticated mixing parameterization. Finally, when comparing all

model results with those from the $180°$-approach flow test only, it was found that this test with the model parameters adopted

from the $0°$-approach flow runs proofed to be not significantly less reliable. This is largely attributed to the simplicity of the

model, as it requires little free parameter tuning.

### 3.5    Urban area within idealized basin

In this simulation, additional model features are combined for a more realistic test case, compared to the previous ones. For

simplicity, this test is quasi 2-D. The third dimension is needed only for a realistic 3-D turbulent mixing, as turbulent mixing

behaves fundamentally different using two dimensions only.

    The domain spans $2880\,\mathrm{m}$ in the lateral direction (x-dimension), $640\,\mathrm{m}$ in the streamwise direction (y-dimension) and $840\,\mathrm{m}$

in the vertical one (z-dimension). The horizontal grid spacing is $20\,\mathrm{m}$ and grid stretching is applied in the vertical dimension.

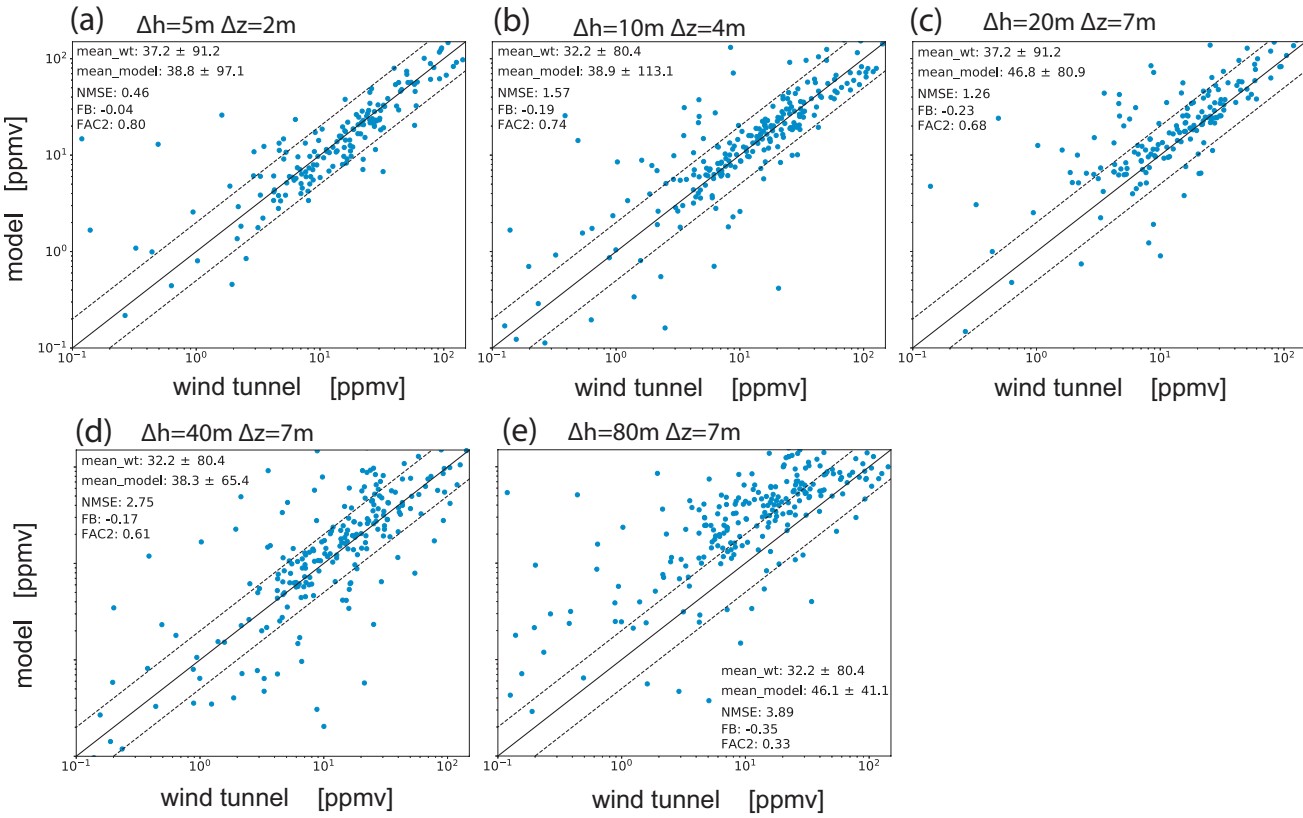

**Figure 14.** Scatter plots of measured vs. modeled concentrations combined for all sources and different grid resolutions. The dotted lines confine the region within a factor of two of measurements.

The grid spacing for the lower most levels is $7\,\mathrm{m}$, and is increased beyond with approaching $50\,\mathrm{m}$ for the upper most levels. An

urban area represented by rectangular buildings is placed at the bottom of an idealized basin whose cross-section is modelled by the following analytic expression for the terrain-height function:

$$h(x) = -\Delta h \exp\left(-\frac{|x|^3}{L^3}\right) \tag{66}$$

$$L = 1\,\mathrm{km}$$

$$\Delta h = 200\,\mathrm{m}$$

The terrain-height function is constant in the streamwise direction, and also the building pattern is repeated in this direction. For all rigid boundaries, including the diffusely discretized buildings, a surface roughness length of $z_0 = 0.1\,\mathrm{m}$ is assumed. The initially neutral stratification of the atmosphere ($\Theta_v = 280\,\mathrm{K}$) is forced towards stable or unstable conditions by prescribing




a constant surface temperature, which, in relation to the initial state of the atmosphere, is cooler or warmer by $5\,\mathrm{K}$. For the heating case, also vertical building walls and roofs have an increased surface temperature, whereas for the cooling case, their
temperature is kept at $\Theta_v = 280\,\mathrm{K}$. The wind field is initialized with an uniformly constant horizontal wind speed of $v = 1\,\mathrm{m\,s^{-1}}$ blowing through the valley. Periodic lateral boundary conditions are used for the y-dimension, and radiation boundary conditions for the x-dimension. A Rayleigh damping layer relaxes the state variables $u$, $v$, $w$ and $\Theta_v$ at the domain top. An area source of a test tracer, which emits at a rate of $1\,\mathrm{u\,s^{-1}}$, is placed at the ground near and between the buildings, mimicking traffic emissions. The boundary conditions for the test tracer are homogeneous-zero Dirichlet on all sides, preventing a re-
entrance of the plume on the upstream domain boundary. For the cooling-case the model was integrated for $18\,\mathrm{h}$ after which an inversion layer was well established. For the heating case, after an integration time of $6\,\mathrm{h}$, no further development of the flow and dispersion pattern occurred.

Figure 15 shows maps of $\Theta_v$ and the tracer concentration fields for both the convective and stable case at the instantaneous times of $6\,\mathrm{h}$ and $18\,\mathrm{h}$, respectively. The fields are averaged along the y-dimension to depict the basin cross section. In the convective
case, 4 pronounced rotors are symmetrically aligned across the basin, with three main near-surface convergence zones: One located directly over the city, and the remaining two on the basin edges. Heat transfer is locally enhanced either by higher surface wind speeds over the basin slopes or through the additional surface area of the buildings within the city. As a result, a pronounced heat island can be observed over the city, which is responsible for the lifting of the air pollution plume originated from the city inside the strong updraught. The air pollution eventually mixes throughout the height range of the simulated
boundary layer. Near-surface air pollution in the city remains comparatively low and concentration gradients are weak. In the stable case, only a shallow boundary layer develops. However, the basin is filled with cooler air and a pronounced inversion layer is present. Below the inversion layer, an erratic shallow circulation pattern with weak horizontal winds is present. Also noticeable are weak katabatic winds blowing down-slope towards the city, where the coldest air gathers. The inversion layer keeps air pollution trapped, and therefore the concentrations increase with advancing simulation time. This phenomenon of
increased air pollution inside city basins is commonly observed during winterly high-pressure periods. While the air pollution within the basin is distributed evenly horizontally it decreases with height, the highest concentration being present at street level in the city.

In this example, it is demonstrated that the effects of a wavy surface orography interact in a complex way with the thermal effects resulting from surface-heat transfer. This is especially true for the convective case, where both the urban heat island effect and the lee effect of the slope have a large influence on the location of the zones of convergence. In any case, the
incorporation of terrain effects from the surroundings is crucial for other more accurate dispersion simulation, which highlights the importance of a holistic simulation approach. From a numerical point of view, the use of a curve-linear grid is advantageous over a standard Cartesian approach, as the vertical grid stretching can be applied just in the same way as for a flat domain. This results in a much lower number of grid boxes for the same near-surface resolution.



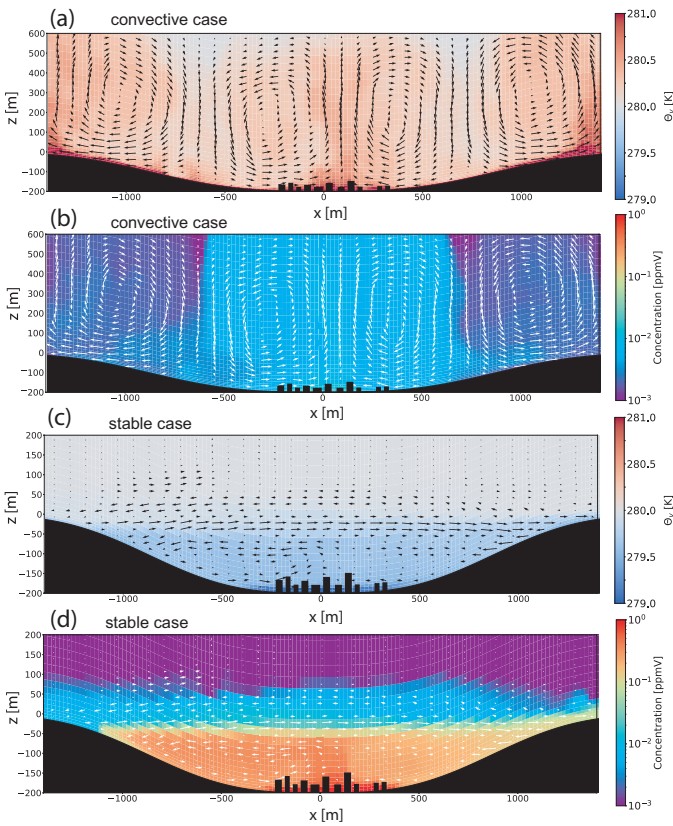

**Figure 15.** Model results of the idealized urban dispersion simulation averaged along the y-axis at 6 h simulation time (convective case) and 18 h simulation time (stable case). The panels (a) and (c) show the virtual potential temperature for a warmer and a cooler surface, which results in an unstable and stable stratification, respectively. Panels (b) and (d) show the implications on air pollution dispersion. To better display the cross-stream circulation pattern in the stable case, only the lower part of the domain is shown and the wind vectors are magnified 10-fold compared to those of the convective case.

## 4 Summary

In this paper, a new large-eddy based dispersion modeling approach for urban application was presented. The model uses diffusive obstacle boundaries in the framework of a finite volume discretization to represent building walls at a wide range of spatial resolutions. Diffusive obstacle boundaries allow for a consistent implementation of buildings in the model code, as they are essentially described by a scalar field for the volume-scaling factors and a vector field for the area-scaling factors. Using these fields to discretize the differential operators, boundary conditions are incorporated naturally and the governing equations are solved for the entire computation grid without requiring to discern different types of grid cells. This permits a straightforward and vectorized implementation of spatial operations. The inherent option for under-resolved diffusive buildings enables





the model to be applied at marginal grid resolutions inaccessible for conventional Cartesian-grid models. The computational
savings can be invested in larger domains to model whole cities and its surroundings. The large-scale terrain influence can be
efficiently represented by curve-linear grids. To benefit from modern hardware architecture, the model is parallelized using a
2D-domain decomposition method, which is sufficient for the expected large grid-aspect ratios of typical boundary-layer appli-
cations. The numerical schemes used are approved and efficient choices. Linear upwind schemes of selectable order of accuracy
with optional limiting are used for advection, a static Smagorinsky model for subgrid turbulence modeling, and multi-stage
higher-order time methods for model integration. The coupling with the mesoscale meteorology can be obtained through dif-
ferent forcing methods using data from a mesoscale host model. A 2-D advection test with a known analytic solution revealed
that the spatial accuracy of the scheme is in the expected range of the design. The sensitivity study with the test-tracer advec-
tion by a potential-flow through an obstacle field demonstrated the robustness of the obstacle discretization up to grid spacings
where the resolution capability of the numerical schemes start to interfere. The model results of the rising thermal experiment
are plausible and similar to those presented in the original studies. The advantages of the model design were demonstrated,
in particular the diffusive obstacle treatment. When compared to data of the "Michelstadt" wind tunnel experiment, the model
simulated reliably complex wind fields and embedded tracer dispersion. As a result, even for spatial resolutions beyond $20\,\mathrm{m}$,
at which buildings can only be represented as increasingly diffuse features, the sensitivity study researching grid-spacing of the
dispersion test showed promising results for a future study with more realistic emission distributions and real mid-sizes cities.
In near future, also the coupling with mesoscale meteorology will be addressed. From previous and accompanying air-quality
studies, simulations with the regional CTM COSMO-MUSCAT are available for different German cities, including Berlin and
Leipzig, for which also comprehensive measurement data are available for model evaluation. In this framework, a promising
application could be a more comprehensive and holistic model evaluation with field data, as additionally to airy monitoring,
mobile measurements may become available for the city of Leipzig. Potential model improvements worth addressing in the
future are the parameterization of air pollution sinks and the implementation of a simple urban atmospheric chemistry. Alterna-
tively, it would be interesting to include diffusive obstacle boundaries in MUSCAT and to investigate whether there is a benefit
in the application on urban air pollution modeling.

*Code and data availability.* The source code of CAIRDIO model version 1.0, utilities for pre- and postprocessing, as well as evaluation data
are accessible in release under the license GPL v3 and later at https://doi.org/10.5281/zenodo.4159497 (Weger et al., 2020).

*Author contributions.* Michael Weger contributed in model development, implementation, evaluation, and paper writing. Oswald Knoth and
Bernd Heinold assisted in model development, paper writing and proof reading.

*Competing interests.* All authors declare that they have no competing interests.




*Acknowledgements.* Wind tunnel data of the "Michelstadt" case were kindly provided by Bernd Leitl and Frank Harms from the Meteorological Institute, University of Hamburg, Germany. The authors gratefully acknowledge the Centre for Information Service and High Performance Computing (Zentrum für Informationsdienste und Hochleistungsrechnen, ZIH) TU Dresden for computing time and their service.

**Appendix A**

| source | mean | wt mean | model NMSE | FB | FAC2 | 180° approach flow | | | | |
|---|---|---|---|---|---|---|---|---|---|---|
| (Δh) | [ppmv] | [ppmv] | | | | S5* (5 m) | 42.4 ±131.9 | 49.4 ±154.7 | 0.49 | -0.15 0.76 |
| | | | | | | S5* (10 m) | 42.4 ±131.9 | 69.8 ±227.2 | 3.19 | -0.49 0.71 |
| **0° approach flow** | | | | | | S5* (20 m) | 42.4 ±131.9 | 45.9 ±118.8 | 0.18 | -0.08 0.68 |
| | | | | | | S5* (40 m) | 42.4 ±131.9 | 43.8 ±87.2 | 2.41 | -0.03 0.65 |
| S2 (5 m) | 16.2 ±22.5 | 14.5 ±18.5 | 0.10 | 0.12 | 0.84 | S5* (80 m) | 42.4 ±131.9 | 48.3 ±55.9 | **7.10** | -0.13 0.50 |
| S2 (10 m) | 16.2 ±22.5 | 13.7 ±17.3 | 0.25 | 0.17 | 0.74 | S6* (5 m) | 54.1 ±126.8 | 57.5 ±127.7 | 0.21 | -0.06 0.78 |
| S2 (20 m) | 16.2 ±22.5 | 8.3 ±9.4 | 1.35 | 0.65 | 0.50 | S6* (10 m) | 54.1 ±126.8 | 58.5 ±155.2 | 0.47 | -0.08 0.89 |
| S2 (40 m) | 16.2 ±22.5 | 17.8 ±23.8 | 0.30 | -0.09 | 0.82 | S6* (20 m) | 54.1 ±126.8 | 59.0 ±136.6 | 1.32 | -0.09 0.78 |
| S2 (80 m) | 16.2 ±22.5 | 29.1 ±32.0 | 0.96 | -0.57 | 0.32 | S6* (40 m) | 54.1 ±126.8 | 53.3 ±94.1 | 2.91 | 0.01 0.76 |
| S4 (5 m) | 4.0 ±3.6 | 4.5 ±4.0 | 0.18 | -0.14 | 0.71 | S6* (80 m) | 54.1 ±126.8 | 41.5 ±33.6 | **6.19** | 0.26 0.54 |
| S4 (10 m) | 4.0 ±3.6 | 4.4 ±3.6 | 0.07 | -0.11 | 1.00 | S7* (5 m) | 37.8 ±63.9 | 37.0 ±70.8 | 0.39 | 0.02 0.86 |
| S4 (20 m) | 4.0 ±3.6 | 5.2 ±4.7 | 0.50 | -0.26 | 0.43 | S7* (10 m) | 37.8 ±63.9 | 44.1 ±72.3 | 0.58 | -0.16 0.79 |
| S4 (40 m) | 4.0 ±3.6 | 6.3 ±5.8 | 0.60 | -0.46 | 0.57 | S7* (20 m) | 37.8 ±63.9 | 53.8 ±89.9 | 1.76 | -0.35 0.63 |
| S4 (80 m) | 4.0 ±3.6 | 18.9 ±12.1 | 1.46 | **-1.30** | **0.0** | S7* (40 m) | 37.8 ±63.9 | 43.3 ±60.0 | 2.20 | -0.14 0.55 |
| S5 (5 m) | 25.7 ±40.8 | 27.1 ±72.8 | 2.58 | -0.05 | 0.48 | S7* (80 m) | 37.8 ±63.9 | 49.9 ±33.8 | 1.66 | -0.28 0.33 |
| S5 (10 m) | 25.7 ±40.8 | 30.1 ±43.0 | 1.02 | -0.16 | 0.52 | S8* (5 m) | 11.4 ±7.8 | 9.6 ±7.1 | 0.05 | 0.18 0.88 |
| S5 (20 m) | 25.7 ±40.8 | 38.5 ±57.9 | 2.11 | -0.40 | 0.67 | S8* (10 m) | 11.4 ±7.8 | 13.5 ±14.1 | 1.06 | -0.16 0.55 |
| S5 (40 m) | 25.7 ±40.8 | 41.1 ±63.8 | 1.52 | -0.48 | 0.62 | S8* (20 m) | 11.4 ±7.8 | 10.6 ±11.2 | 1.19 | 0.08 0.42 |
| S5 (80 m) | 25.7 ±40.8 | 55.6 ±56.2 | 2.30 | **-0.73** | **0.24** | S8* (40 m) | 11.4 ±7.8 | 31.3 ±40.8 | 4.06 | **-0.93** 0.30 |
| | | | | | | S8* (80 m) | 11.4 ±7.8 | 58.4 ±40.3 | 1.74 | **-1.35** **0.03** |

**Table A1.** Statistical results of all dispersion simulations performed in the "Michelstadt" wind tunnel simulation study. The cases superscripted with a star were carried out using the reverse approach-flow direction of 180°. Missed acceptance criteria are highlighted in bold font.



| source ($\Delta h$) | mean [ppmv] | wt mean [ppmv] | model NMSE | FB | FAC2 |
|---|---|---|---|---|---|
| All (5 m) | 32.2 ±80.4 | 33.0 ±89.5 | 0.54 | -0.02 | 0.80 |
| All (10 m) | 32.2 ±80.4 | 38.9 ±113.1 | 1.57 | -0.19 | 0.74 |
| All (20 m) | 32.2 ±80.4 | 38.5 ±89.5 | 1.72 | -0.18 | 0.61 |
| All (40 m) | 32.2 ±80.4 | 38.3 ±65.4 | 2.75 | -0.17 | 0.61 |
| All (80 m) | 32.2 ±80.4 | 46.1 ±41.1 | 3.89 | -0.35 | 0.33 |
| All* (5 m) | 37.2 ±91.2 | 38.5 ±100.2 | 0.40 | -0.03 | 0.83 |
| All* (10 m) | 37.2 ±91.2 | 46.2 ±129.5 | 1.48 | -0.22 | 0.75 |
| All* (20 m) | 37.2 ±91.2 | 45.8 ±100.7 | 1.50 | -0.21 | 0.63 |
| All* (40 m) | 37.2 ±91.2 | 43.2 ±71.2 | 2.72 | -0.15 | 0.56 |
| All* (80 m) | 37.2 ±91.2 | 49.4 ±40.1 | 3.97 | -0.28 | 0.35 |

**Table A2.** Statistical results derived from the combined data of all simulated sources at the given spatial resolution. The cases superscripted with a star are the combined results of the 180 ° approach-flow cases only.

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
