# Peer review of "An urban large-eddy-based dispersion model for marginal grid resolutions: CAIRDIO v1.0"

_Geoscientific Model Development, 2020_

## Referee Comment (RC1) · Anonymous Referee #1 · 1 Dec 2020

Dear authors,

This manuscript represents a novel large-eddy simulation (LES) model based on the diffusive interface approach. The main interest is in urban pollutant dispersion on the mesoscale. The manuscript contains an exhaustive description of the model equations as well as results on modelling different phenomena on varying scales. The main objective in model development has been on reducing the computational costs, which in general are high for urban LES studies.

Overall, the manuscript is well prepared and the representation scientifically sound. However, I have some major comments related to the readability and aim of the

manuscript:

1. The manuscript is rather long, which was already pointed out by the Editor. In my opinion, one solution for this could be to divide the manuscript in two, e.g.: 1) a more simple and concise model description manuscript with a simple evaluation study and 2) a manuscript presenting some atmospheric applications and sensitivity of the model. Now the model evaluation (Section 3) contains five different studies. Yet, only the comparison to the Michelstadt wind tunnel experiments (Section 3.4) represents a model evaluation. Section 3.2, instead, illustrate the model sensitivity. The rest of the studies illustrate the applicability of the model, but it is impossible to say how well does the model perform. Also, I think the annulus advection test might not be the most suitable one for a geoscientific journal.

2. The model description part of the manuscript is rather exhaustive and sometimes some variables are not introduced in the close vicinity of the equation. Please check these. You could also think if you could come up with some illustrative figures for this section.

3. The objective to limit the computational costs of an LES model is very important. However, one should keep in mind what is the aim of the application. For instance, to resolve the flow in urban street canyons and courtyards, a spatial resolution of $\sim$1 m is needed in order to directly resolve most of the energy and keep it LES. This should be emphasized in the manuscript. Furthermore, I find the first line of the abstract misleading. High spatial resolutions are needed to ensure accuracy in urban LES and to keep the amount of energy resolved by SGS terms small.

4. The language needs revision.

Additionally, find below some minor comments. P indicates page and L line.

- P1 title: "large-eddy-based" does not mean anything in my opinion. I would change it to "large-eddy-simulation-based"

- P1 L5: Why not also vertical resolution?

- P1 L13-14: I should be stated here whether the evaluation was successful or not.

- P1 L19: I do not think you can say that the PBL mixing processes can be well parametrised for urban areas. . .

- P1 L20-23: There is something missing in this phrase. Now it indicates that "providing more representative forecasts for individual locations" would be a research purpose.

- P2 L55: add "e.g." for the reference to Maronga et al. (2019), since the preceding statement about LES is not initially from this specific publication

- P2 L59-60: To conduct obstacle-resolving LES in urban areas, a spatial resolution of ∼1 m is needed (see e.g. Xie, Z., Castro, I.P. LES and RANS for Turbulent Flow over Arrays of Wall-Mounted Obstacles. Flow Turbulence Combust 76, 291 (2006). https://doi.org/10.1007/s10494-006-9018-6). Also, "less than 10 m to 20 m" sounds weird.

- P4 Eq. 1: Introduce u

- P5 Eq. 4: Introduce theta (yes, theta_v has been introduced)

- P6 L156: Typically, the continuity equation is referred to instead of the "divergence-free criterion"

- P6 L163-164: the notation "z=const" is unclear. Maybe simply "z is constant".

- P6 Eq. 13: Introduce A and V.

- P7 Eq. 15: the notation f was already applied for the Coriolis term

- P7 L189: z and x should be in italics

- P9 L225: I would but "mod 2" inside a parenthesis

- P9 L226: remove "it is" before "r = " and introduce j

- P10 Eq. 24: what does "Limods" indicate?

- P11 Eq. 28: It is not clear how this is derived

- P11 L281: "a first-order accurate in time Euler method" should be rewritten

- P11 L287: solve –> solver

- P11 L288: st and nd in 1st and 2nd should not be in italics

- P12 L294: is G used somewhere?

- P17 Eqs. 42-43: The transition from the Eq. 42 to Eq. 43 is very unclear.

- P25 L565: If you want to evaluate the model, it would be a good idea to plot also the results of the original study by Wicker and Skamarock (1998).

- Section 3.4: The description of the wind tunnel experiment can be shortened as it has already been published in Baumann-Stanzer et al. (2015).

- P28 L642-644: Give a reference for this phrase.

- P30 L646: If only the continuously emitting mode is used, why to mention the other at all here?

- P30 L650: The acceptance limits are originally from Hanna, S. and Chang, J. (2012): Acceptance criteria for urban dispersion model evaluation, Meteorol. Atmos. Phys., 116, 133–146, https://doi.org/10.1007/s00703-011-0177-1

- P30 L663: remove "of" after "Instead"

- P34 L718: is "u/s" the correct unit here?

- P34 L729: remove "range"

---

## Referee Comment (RC2) · Anonymous Referee #2 · 20 Dec 2020

General comments

The paper describes a newly developed LES-based dispersion model suitable on the urban scale to simulate boundary layer flow and air pollutants. Herein, the authors use a known approach from two-phase modeling (diffusive obstacle boundaries) and apply it to an urban LES model for the first time. It is shown that this approach is computationally efficient even when grid spacings are in the order of the buildings itself. Overall, the content is presented well and in a scientifically sound manner.

Specific comments

I suggest that the paper be published in GMD after the authors have addressed the

following points:

1. Some sections of the manuscript are exhaustive and described in great detail. If the authors decide to keep everything within one paper, I suggest to shorten several parts of the paper to increase the overall readability.

2. The "Michelstadt" wind tunnel experiment is a very nice example for model evaluation. However, the other parts of Section 3 are rather numerical sensitivity and convergence tests. Therefore, this section should be divided into two separate sections.

3. Some technical parts of the model itself should be mentioned or explained. For example, why is it called CAIRDIO (if this is abbreviation, what does it stand for?). In which programming language is the code written? Which libraries are used? Also, since the authors argue that the main benefit is the increased computational efficiency due to the diffusive interface approach, some kind of scaling analysis for a varying number of CPUs (or nodes) to test the parallelization would surely be of interest.

4. For most of the figures, the font size (especially for the axis labels) needs to be increased.

5. The authors should spend at least one more iteration on checking language and grammar as well as formatting inline equations.

Technical corrections

P1 L1: Instead of "accurate numerical models" I would write "numerical models dedicated to accurately simulate".

P1 L6: "like e.g. the" -> "like, e.g., the"

P1 L12: What does "mid-sized" mean?

P2 L57: The term "terra-incognita" already existed before.

P2 L59: "stringend" -> "stringent"

P3 L62: "e.g.,"

P3 L67-68: What do you mean by "more holistic simulations"?

P4 Eq. 1: Introduce rho

P4 L117: Avoid putting a variable name at the beginning of a sentence.

P5 L136: "Computation grid" -> "Computational grid"

P10 L242-243: The superscripts "th" and "rd" should be in text-mode, not math-mode.

P10 L258: "while in" -> "while within"

P11 L268: Wrong citation style

P11 L285+288: "2nd" "3rd" etc. (see above). Please check all further appearances in the whole manuscript.

P13 Fig. 2 caption: Which exactly are the different resolutions for a) – f)?

P30 L648: NMSE and FB have already been introduced.

P30 L671: This is one example where inline equations are not properly formatted ("NMSE= 0.10"). Please check all occurrences.

P31 L679: delete "ever"

P31 L684+685: "Tab." -> "Table"

P31 L693: "over-estimates" -> "overestimates"

P32 L702+703: "x", "y" and "z" should be in math-mode. Please check for the whole manuscript.

P33 L704: "lower most" -> "lowermost", "upper most" -> "uppermost"

P34 L715: "an uniformly" -> "a uniformly"

P34 L715+716: Correctly format inline equations to avoid linebreaks within them

P34 L718: What unit is "u sˆ-1"?

P34 L735: Increased air pollution is also observed in stable boundary layers during nighttime (not only winterly high-pressure periods).

P36 L771-773: I would rewrite the sentence in the following way: "In this framework, a promising application could be a more comprehensive and holistic model evaluation with field data, as mobile measurements could be available for the city of Leipzig in addition to air monitoring."

———————————————————

---

## Author Comment (AC1) · 23 Jan 2021

Firstly, we would like to thank the Referees for their valuable comments and suggestions, which helped us to improve the quality and readability of the manuscript. In this document, we respond to the specific comments of the Referees, starting with Referee 1. For all responses to the minor comments, we refer to the additionally supplemented pdf, where Referee comments are highlighted in red.

Response to comments of Referee 1

Specific comments

"The manuscript is rather long, which was already pointed out by the Editor. In my opinion, one solution for this could be to divide the manuscript in two, e.g.: 1) a more simple and concise model description manuscript with a simple evaluation study and 2) a manuscript presenting some atmospheric applications and sensitivity of the model. Now the model evaluation (Section 3) contains five different studies. Yet, only the comparison to the Michelstadt wind tunnel experiments (Section 3.4) represents a model evaluation. Section 3.2, instead, illustrate the model sensitivity. The rest of the studies illustrate the applicability of the model, but it is impossible to say how well does the model perform. Also, I think the annulus advection test might not be the most suitable one for a geoscientific journal."

We agree therein that parts of the manuscript were too extensive and some of the studies would better fit in a separate manuscript dedicated to model application and sensitivity. We tried to solve this issue by removing some of the studies and by shortening the model description. The annulus convergence test, as already pointed out, is not the most suitable one for this manuscript, so, we removed it. We also agree that the study with the idealized city basin is more of an illustrative example and does not really contribute to a better understanding on how well the model performs. This study will be replaced by an application study on a real city in the aforementioned separate manuscript. The rest of the studies we decided to keep for now and we collected them under Section 3 "numerical studies", which also contains a third study concerning parallel scalability, as suggested by Referee 2 (see comments below). If the manuscript is still too long, we can remove this section completely and move the scalability test to the paragraph "programming language" of the model description. Section 4 now contains the evaluation study with the wind-tunnel experiment.

2. "The model description part of the manuscript is rather exhaustive and sometimes some variables are not introduced in the close vicinity of the equation. Please check these. You could also think if you could come up with some illustrative figures for this section."

We shortened the model description part where possible and reorganized some sub-sections to improve the overall structure. For example, the description of the advection scheme was rewritten in a much more compact form, as it previously contained a lot of text-book knowledge, which can be looked-up in other papers. We supplemented it with an illustrative Figure to demonstrate the reconstruction near diffuse obstacle boundaries. Non-essential parts of the pressure-solver description were removed for similar reasons, as there are many publications about multigrid algorithms. Most of the other subsections were shortened too. However, a few sentences were added to mention the programming language and the packages we used, as suggested by Referee 2. We further checked the manuscript for variables not introduced in the vicinity where they are used. In conclusion, the model description part is now shorter, and hopefully better to read.

3. "The objective to limit the computational costs of an LES model is very important. However, one should keep in mind what is the aim of the application. For instance, to resolve the flow in urban street canyons and courtyards, a spatial resolution of âĹij1 m is needed in order to directly resolve most of the energy and keep it LES. This should be emphasized in the manuscript. Furthermore, I find the first line of the abstract misleading. High spatial resolutions are needed to ensure accuracy in urban LES and to keep the amount of energy resolved by SGS terms small."

This is indeed an important remark which was not emphasized enough in the manuscript. We also thankfully incorporated the study of Xie and Castro (2006) which researched grid-sensitivity of LES models. In the introduction we now clearly state, that for an LES-model to be fully LES also within the urban boundary layer a spatial resolution of ∼1m is needed. Our application with diffusive buildings can therefore be interpreted as a hybride approach (or partly under-resolved LES) which still works well for the purpose of urban air-quality modeling on a larger scale (see also the cited study by Wolf et. al (2020)). However, if one researches the detailed wind field surrounding buildings, then clearly more spatial resolution is needed which comes always

at its computational costs. Hence, our approach cannot make such simulations more efficient. But it can make urban microscale dispersion simulations cheaper, as the diffusive obstacle approach can shift the technical limits toward the coarser mesoscale. We carefully checked the use of the word "accuracy" in the manuscript as it shall not refer to the accuracy of LES models.

4. "The language needs revision."

We revised the language and checked the whole manuscript for grammar and punctuation.

Response to comments of Referee 2

Specific comments

1." Some sections of the manuscript are exhaustive and described in great detail. If the authors decide to keep everything within one paper, I suggest to shorten several parts of the paper to increase the overall readability."

This was already pointed out by Referee 1, and we therefore kindly refer to the first major comment.

2. "The "Michelstadt" wind tunnel experiment is a very nice example for model evaluation. However, the other parts of Section 3 are rather numerical sensitivity and convergence tests. Therefore, this section should be divided into two separate sections."

We followed this suggestion and put the Michelstadt wind tunnel experiment in Section 4. Section 3 is used for the three numerical tests, now consisting of the advection test with the circular obstacles, the rising thermal experiment, and the parallel scalability test. However, if the Reviewer suggests so, we can remove Section 3 entirely and move the scalability test to the model description part.

3. "Some technical parts of the model itself should be mentioned or explained. For example, why is it called CAIRDIO (if this is abbreviation, what does it stand for?). In

which programming language is the code written? Which libraries are used? Also, since the authors argue that the main benefit is the increased computational efficiency due to the diffusive interface approach, some kind of scaling analysis for a varying number of CPUs (or nodes) to test the parallelization would surely be of interest."

The full model name is now contained in the abstract and introduction. The code is written in Python and we added a short paragraph where we also mention the Python libraries we used. A parallel scaling test of the model is indeed very interesting, so we followed your suggestion and included such a test. We tested strong scalability for a range of 1- 400 CPU cores and a constant test problem.

4. "For most of the figures, the font size (especially for the axis labels) needs to be increased."

The font size of most figures was increased, especially of single-column figures.

5. "The authors should spend at least one more iteration on checking language and grammar as well as formatting inline equations."

We revised language, grammar and equation formatting.

Minor comments

See supplementary pdf-file

Please also note the supplement to this comment:
https://gmd.copernicus.org/preprints/gmd-2020-313/gmd-2020-313-AC1-supplement.pdf

**Supplement:**

**Athor's responce to Referee comments**

Firstly, we would like to thank the Referees for their valuable comments and suggestions, which helped us to improve the quality and readability of the manuscript. In this document, we respond to the individual comments of both Referees. Referee comments are highlighted in red.

**Response to comments of Referee 1**

**Major comments**

The manuscript is rather long, which was already pointed out by the Editor. In my opinion, one solution for this could be to divide the manuscript in two, e.g.: 1) a more simple and concise model description manuscript with a simple evaluation study and 2) a manuscript presenting some atmospheric applications and sensitivity of the model. Now the model evaluation (Section 3) contains five different studies. Yet, only the comparison to the Michelstadt wind tunnel experiments (Section 3.4) represents a model evaluation. Section 3.2, instead, illustrate the model sensitivity. The rest of the studies illustrate the applicability of the model, but it is impossible to say how well does the model perform. Also, I think the annulus advection test might not be the most suitable one for a geoscientific journal.

We agree therein that parts of the manuscript were too extensive and some of the studies would better fit in a separate manuscript dedicated to model application and sensitivity. We tried to solve this issue by removing some of the studies and by shortening the model description. The annulus convergence test, as already pointed out, is not the most suitable one for this manuscript, so, we removed it. We also agree that the study with the idealized city basin is more of an illustrative example and does not really contribute to a better understanding on how well the model performs. This study will be replaced by an application study on a real city in the aforementioned separate manuscript. The rest of the studies we decided to keep for now and we collected them under Section 3 "numerical studies", which also contains a third study concerning parallel scalability, as suggested by Referee 2 (see comments below). If the manuscript is still too long, we can remove this section completely and move the scalability test to the paragraph "programming language" of the model description. Section 4 now contains the evaluation study with the wind-tunnel experiment.

2. The model description part of the manuscript is rather exhaustive and sometimes some variables are not introduced in the close vicinity of the equation. Please check these. You could also think if you could come up with some illustrative figures for this section.

We shortened the model description part where possible and reorganized some subsections to improve the overall structure. For example, the description of the advection scheme was rewritten in a much more compact form, as it previously contained a lot of text-book knowledge, which can be looked-up in other papers. We supplemented it with an illustrative Figure to demonstrate the reconstruction near diffuse obstacle boundaries. Non-essential parts of the pressure-solver description were removed for similar reasons, as there are many publications about multigrid algorithms. Most of the other subsections were shortened too. However, a few sentences were added to mention the programming language and the packages we used, as suggested by Referee 2. We further checked the manuscript for variables not introduced in the vicinity where they are used. In conclusion, the model description part is now shorter, and hopefully better to read.

3. The objective to limit the computational costs of an LES model is very important. However, one should keep in mind what is the aim of the application. For instance, to resolve the flow in urban street canyons and courtyards, a spatial resolution of  $\sim 1$  m is needed in order to directly resolve most of the energy and keep it LES. This should be emphasized in the manuscript. Furthermore, I find the first line of the abstract misleading. High spatial resolutions are needed to ensure accuracy in urban LES and to keep the amount of energy resolved by SGS terms small.

This is indeed an important remark which was not emphasized enough in the manuscript. We also thankfully incorporated the study of Xie and Castro (2006) which researched grid-sensitivity of LES models. In the introduction we now clearly state, that for an LES-model to be fully LES also within the urban boundary layer a spatial resolution of ~1m is needed. Our application with diffusive buildings can therefore be interpreted as a hybride approach (or partly under-resolved LES) which still works well for the purpose of urban air-quality modeling on a larger scale (see also the cited study by Wolf et. al (2020)). However, if one researches the detailed wind field surrounding buildings, then clearly more spatial resolution is needed which comes always at its computational costs. Hence, our approach cannot make such simulations more efficient. But it can make urban microscale dispersion simulations cheaper, as the diffusive obstacle approach can shift the technical limits toward the coarser mesoscale. We carefully checked the use of the word "accuracy" in the manuscript as it shall not refer to the accuracy of LES models.

**4. The language needs revision.**

We revised the language and checked the whole manuscript for grammar and punctuation.

**Minor comments**

P1 title: "large-eddy-based" does not mean anything in my opinion. I would change it to "large-eddysimulation-based".

We adopted this suggestion.

**P1 L5: Why not also vertical resolution?**

It is true, that also an adequate vertical grid resolution is required. In this respect, the sentence was changed to "The major reason for the inflated numerical costs is the required spatial resolution to meaningfully apply the obstacle discretization". We need, however, to emphasize that increasing vertical grid spacing is not as computationally demanding as increasing horizontal grid spacing for two reasons: Firstly, the amount of additional computation scales only linearly, as opposed to quadratic for an increase in both horizontal dimension sizes. Secondly, the CFL-criterion is imposed by the horizontal grid spacing when the wind speed is dominated by the horizontal components. This will be even the case for grids with high aspect ratios, because the vertical wind speed is averaged over a larger horizontal area and therefore much smaller in magnitude. For this reason, also numerical weather prediction models with high grid-aspect ratios can be efficiently integrated, despite using quite a high vertical resolution near the ground.

**P1 L13-14: It should be stated here whether the evaluation was successful or not.**

We extended the abstract with a few lines on the outcome of the evaluation study.

**- P1 L19: I do not think you can say that the PBL mixing processes can be well parametrised for urban areas. . .**

The sentence likely suggests that PBL mixing processes for urban areas can be well parametrized. This would indeed be misleading, as parameterizations can never be as accurate as the explicit representation of processes. Also parameterized mixing is only a contribution to the total mixing. To make this clear, we changed the phrase to:

"On the one hand, even though PBL mixing processes are often parameterized to a large extend, the parameterizations themselves must rely on a sound physical basis, for which detailed large-eddy simulations (LES) can be consulted (Noh and Raasch, 2003; Kanda et al., 2013)."

**P1 L20-23: There is something missing in this phrase. Now it indicates that "providing more representative forecasts for individual locations" would be a research purpose.**

This was rearranged to make the statement clear:

"Direct benefits of more detailed numerical simulations include an increased ability to produce more representative air-quality forecasts for individual locations (Carlino et al., 2016) and the provision of high-resolution 4-dimensional data for research purposes, like, e.g., source attribution (Fernández et al., 2019) and exposure risk assessment (Chang, 2016)."

- P2 L55: add "e.g." for the reference to Maronga et al. (2019), since the preceding statement about LES is not initially from this specific publication

Added.

- P2 L59-60: To conduct obstacle-resolving LES in urban areas, a spatial resolution of ~1 m is needed (see e.g. Xie, Z., Castro, I.P. LES and RANS for Turbulent Flow over Arrays of Wall-Mounted Obstacles. Flow Turbulence Combust 76, 291 (2006). https://doi.org/10.1007/s10494-006-9018-6). Also, "less than 10 m to 20 m" sounds weird.

We added a few lines in the introduction to clarify that for a truly obstacle resolving LES such a high resolution is indeed required (along with the given citation).

**P4 Eq. 1: Introduce u**

It is now introduced after Eq. 1 along with the missing introduction of the air density.

- P5 Eq. 4: Introduce theta (yes, theta\_v has been introduced)

It is also now introduced. We mistakenly wrote theta\_v, but the transport equation is for theta.

- P6 L163-164: the notation "z=const" is unclear. Maybe simply "z is constant".

This was now replaced with:

"However, the horizontal averaging of  $\Theta v$  is carried out on z-isosurfaces. Therefore,  $\Theta v$  is remapped to an auxiliary vertical grid and the calculated tendency is remapped back to the computational grid."

- P6 Eq. 13: Introduce A and V.

We introduced the grid-cell volume and the total surface area over which are integrated. A and V serve as formal integration variables here and are now mentioned as such.

-P7 Eq. 15: the notation f was already applied for the Coriolis term

We use now a capital letter for the flux.

- P7 L189: z and x should be in italics

Corrected here and elsewhere in the manuscript.

- P9 L225: I would but "mod 2" inside a parenthesis

We changed it to "mod(n, 2)".

-P9 L226: remove "it is" before "r = " and introduce j

The manuscript does not contain this formula anymore.

- P10 Eq. 24: what does "Limods" indicate?

Also Eq. 24 can be explained in a single sentence and was removed.

- P11 Eq. 28: It is not clear how this is derived

Now we replaced Eq. 28 with the implicit time step to get ut1 by integrating the neglected pressure tendency. The divergence operator is then applied on Eq.28 to give the pressure Poisson equation.

-L281: "a first-order accurate in time Euler method" should be rewritten

It was rewritten to: "Equation 30 is only first-order accurate in time."

- P11 L287: solve -> solver

Corrected.

- P11 L288: st and nd in 1st and 2nd should not be in italics

Corrected here and elsewhere in the manuscript.

- P12 L294: is G used somewhere?

It is not directly used in an Equation. Therefor it was removed.

- P17 Eqs. 42-43: The transition from the Eq. 42 to Eq. 43 is very unclear.

It should now be better understandable as some information was indeed missing.

- P25 L565: If you want to evaluate the model, it would be a good idea to plot also the results of the original study by Wicker and Skamarock (1998).

We agree. We got the permission from AMS to include the original plot in our manuscript.

- Section 3.4: The description of the wind tunnel experiment can be shortened as it has already been published in Baumann-Stanzer et al. (2015).

We shortened the paragraph.

- P28 L642-644: Give a reference for this phrase.

We can also reuse here your suggested reference Xie and Castro (2006) for the introduction. We also made clear that spatial averaging over re-circulation zones results in a lower magnitude of wind speeds compared to measurements.

- P30 L646: If only the continuously emitting mode is used, why to mention the other at all here?

It is now removed, as indeed we tested the continuously emitting mode.

- P30 L650: The acceptance limits are originally from Hanna, S. and Chang, J. (2012): Acceptance criteria for urban dispersion model evaluation, Meteorol. Atmos. Phys., 116, 133–146, https://doi.org/10.1007/s00703-011-0177-1

We were not aware of this and thank the Reviewer for the correct citation.

- P30 L663: remove "of" after "Instead"

Corrected.

- P34 L718: is "u/s" the correct unit here?

We removed this simulation (see comments above).

- P34 L729: remove "range"

See comments above.

**Response to comments of Referee 2**

**Specific comments**

1. Some sections of the manuscript are exhaustive and described in great detail. If the authors decide to keep everything within one paper, I suggest to shorten several parts of the paper to increase the overall readability.

This was already pointed out by Referee 1, and we therefore kindly refer to the first major comment.

2. The "Michelstadt" wind tunnel experiment is a very nice example for model evaluation. However, the other parts of Section 3 are rather numerical sensitivity and convergence tests. Therefore, this section should be divided into two separate sections.

We followed this suggestion and put the Michelstadt wind tunnel experiment in Section 4. Section 3 is used for the three numerical tests, now consisting of the advection test with the circular obstacles, the rising thermal experiment, and the parallel scalability test. However, if the Reviewer suggests so, we can remove Section 3 entirely and move the scalability test to the model description part.

3. Some technical parts of the model itself should be mentioned or explained. For example, why is it called CAIRDIO (if this is abbreviation, what does it stand for?). In which programming language is the code written? Which libraries are used? Also, since the authors argue that the main benefit is the increased computational efficiency due to the diffusive interface approach, some kind of scaling analysis for a varying number of CPUs (or nodes) to test the parallelization would surely be of interest.

The full model name is now contained in the abstract and introduction. The code is written in Python and we added a short paragraph where we also mention the Python libraries we used. A parallel scaling test of the model is indeed very interesting, so we followed your suggestion and included such a test. We tested strong scalability for a range of 1- 400 CPU cores and a constant test problem.

4. For most of the figures, the font size (especially for the axis labels) needs to be increased.

The font size of many figures was increased, especially of single-column figures.

5. The authors should spend at least one more iteration on checking language and grammar as well as formatting inline equations.

We revised language, grammar and equation formatting.

**Minor comments**

P1 L1: Instead of "accurate numerical models" I would write "numerical models dedicated to accurately simulate".

We agree that your suggestion is more appropriate.

P1 L6: "like e.g. the" -> "like, e.g., the"

Corrected.

P1 L12: What does "mid-sized" mean?

Agreed, without definition, it makes little sense to use the term "mid-sized". We wanted to exclude large metropoles from our statement. Maybe the term "city" alone would be informative enough.

P2 L57: The term "terra-incognita" already existed before.

We changed the wording to: "...for which reason Haupt et al. (2019) coined the key word of "terraincognita" to refer to the problem."

P2 L59: "stringend" -> "stringent"

This sentence was replaced.

P3 L62: "e.g.,"

Also this sentence was replaced.

P3 L67-68: What do you mean by "more holistic simulations"?

It should mean more comprehensive simulations. So we use the word "comprehensive" instead of "holistic".

**P4 Eq. 1: Introduce rho**

Rho is now introduced as density.

P4 L117: Avoid putting a variable name at the beginning of a sentence.

We rearranged the sentence to:

"The sum of external body forces b contains the gravitational force and inertial forces resulting in a rotating frame of reference."

P5 L136: "Computation grid" -> "Computational grid"

Corrected.

P10 L242-243: The superscripts "th" and "rd" should be in text-mode, not math-mode.

This and other occurrences in the manuscript are corrected.

P10 L258: "while in" -> "while within"

This sentence was removed.

P11 L268: Wrong citation style

Corrected.

P11 L285+288: "2nd" "3rd" etc. (see above). Please check all further appearances in the whole manuscript.

Corrected (see above).

**P13 Fig. 2 caption: Which exactly are the different resolutions for a) - f?**

The resolution of each grid is now included in the caption.

P30 L648: NMSE and FB have already been introduced.

We overlooked this. Now it reads:

"As proxy for the quality of model results in comparison to the measurements, NMSE, FB and additionally the fraction of within factor 2 (FAC2) are calculated."

P30 L671: This is one example where inline equations are not properly formatted ("NMSE= 0.10"). Please check all occurrences.

Corrected.

P31 L679: delete "ever"

Deleted.

P31 L684+685: "Tab." -> "Table"

Corrected.

P31 L693: "over-estimates" -> "overestimates"

Corrected.

P32 L702+703: "x", "y" and "z" should be in math-mode. Please check for the whole manuscript.

This and all other occurrences are corrected.

P33 L704: "lower most" -> "lowermost", "upper most" -> "uppermost"

P34 L715: "an uniformly" -> "a uniformly"

P34 L715+716: Correctly format inline equations to avoid linebreaks within them

P34 L718: What unit is "u s-1"?

P34 L735: Increased air pollution is also observed in stable boundary layers during nighttime (not only winterly high-pressure periods).

This study was removed from the manuscript.

P36 L771-773: I would rewrite the sentence in the following way: "In this framework, a promising application could be a more comprehensive and holistic model evaluation with field data, as mobile measurements could be available for the city of Leipzig in addition to air monitoring."

We rewrote the sentence in a similar way to your suggestion:

"In this framework, a promising application could be a more comprehensive and holistic model evaluation with field data, as mobile measurements are available for the city of Leipzig in addition to operational air monitoring.

---

## Author Response (AR2)

**Author's response to Editor comments**

(1) Python version 2.7 has been deprecated. Typically, migrating to Python 3 is painless, so I suggest you upgrade your code to work on Python 3 as well.

The migration to Python 3 was already on our to-do list. The updated code is now available at Zenodo: (https://doi.org/10.5281/zenodo.4486984). We also updated respective DIO in the manuscript.

(2) There are precompiled libraries (e.g. MODEL_SRC/sparse_approximate_inverse.so) that cannot be used on other operating systems (like Mac OS X). Please provide a code base that allows to recompile those so the code can be used.

The description on how to precompile some code snippets was still missing in the Readme file. We included this part. In order to do so, the Python package Cython needs to be installed first. Afterwards, please enter the folder "./MODEL_SRC/" and execute the script "compile_clibs.sh". This procedure should work also on MAC OS X.

(3) Your code base on Zenodo contains git folders referencing your repository, you might want to remove these.

Thank you for this hint. We removed superfluous folders and files.

We hope that with these additional instructions it is possible to painlessly run a test example of our code.

In Figure 2a) axis labels were still missing. We resolved that. There are no further changes to the previously accepted manuscript.